# Recurrent Complex-Weighted Autoencoders for Unsupervised Object Discovery

**Anand Gopalakrishnan**[1*] **Aleksandar Stanić**[2†]
**Jürgen Schmidhuber**[1,3]   **Michael Curtis Mozer**[2]
[1]The Swiss AI Lab, IDSIA, USI & SUPSI, Lugano, Switzerland
[2]Google DeepMind
[3]AI Initiative, KAUST, Thuwal, Saudi Arabia

## Abstract

Current state-of-the-art synchrony-based models encode object bindings with complex-valued activations and compute with real-valued weights in feedforward architectures. We argue for the computational advantages of a recurrent architecture with complex-valued weights. We propose a fully convolutional autoencoder, *SynCx*, that performs iterative constraint satisfaction: at each iteration, a hidden layer bottleneck encodes statistically regular configurations of features in particular phase relationships; over iterations, local constraints propagate and the model converges to a globally consistent configuration of phase assignments. Binding is achieved simply by the matrix-vector product operation between complex-valued weights and activations, without the need for additional mechanisms that have been incorporated into current synchrony-based models. SynCx outperforms or is strongly competitive with current models for unsupervised object discovery. SynCx also avoids certain systematic grouping errors of current models, such as the inability to separate similarly colored objects without additional supervision. [3]

## 1 Introduction

When shown arrays of simple visual elements, people have a natural tendency to perceive the arrays in disjoint groups based on their color, shape, spatial configuration, etc. Grouping behavior was first studied and systematized by Gestalt psychologists [1–3] who proposed a set of principles according to which perception operates, e.g., grouping by *proximity*, *similarity*, *closure*, *good continuation* and *common fate*. Although these Gestalt principles—also called laws—were assumed to be innate, statistics of the environment may explain some forms of grouping [4], and people rapidly learn new grouping principles when they are exposed to novel environments [5]. Grouping abilities are acquired early in an infant's developmental cycle [6] and serve as a building block in the human ability to form concepts [7], build abstractions and categorize [8], and reason about the physical world [9].

Perceptual grouping has been modeled in deep learning under the guise of *object-centric learning*. (See Greff et al. [10] for an overview.) Object-centric models discover modular and compositional representations that can facilitate stronger generalization and relational reasoning capabilities on downstream visual tasks such as question answering [11], game playing [12–14] and robotics [15–17].

Both human and AI object-centric learning aim to solve the *binding problem* [18, 19]—determining how to integrate visual information in a flexible, dynamic manner to form unified wholes. The predominant design strategy for implementing binding in deep learning systems to achieve perceptual

---

*Correspondence to `anand@idsia.ch`
†Work done at IDSIA.

[3]Official code repository: `https://github.com/agopal42/syncx`

38th Conference on Neural Information Processing Systems (NeurIPS 2024).

grouping is to maintain a set of latent activation vectors (*slots*), each of which encodes features of just one object. Slot-based models differ from one another in the procedures used to partition information from the inputs to each slot [10]. Far less investigation has gone into an alternative paradigm for perceptual grouping based on synchrony. In this paradigm, bindings between features are expressed in terms of the relative phases of complex-valued neural activities. The earliest demonstration of a synchrony-based model in neural networks focused on a supervised setting with teacher-specified target phases for hard-coded one-hot features of contour types [20], although consideration was given to an unsupervised extension obtained by phase clustering [21]. Recent synchrony-based models [22–25] using complex-valued activations to learn suitable features have made progress in unsupervised grouping performance on synthetic and more naturalistic scenes.

However, current state-of-the-art unsupervised synchrony-based models such as CAE ([23]), CtCAE ([24]), and RF ([25]) have a number of limitations. They employ real-valued weights to process complex-valued activations by weight sharing across the real and imaginary components. As a result, the models do not exploit the constructive or destructive interference of complex-valued activations that occur naturally via multiplication and addition operations (cf., [26]). And the models require use of additional inductive biases such as gating mechanisms ($\chi$-binding [22, 23, 25, 24]) to implement binding. Further, it remains unclear how $\chi$-binding mechanism works in more general scenarios where the complex-valued activations are not exactly out-of-phase or norms of weights and features do not satisfy certain conditions as noted by Löwe et al. [27]. Cosine binding [27] was proposed as a more computationally motivated and interpretable alternative that easily handles such scenarios. However, this model continues to use real-valued weights to process complex-valued activations. Therefore, it uses cosine distance between inputs and intermediate outputs to implement binding which has a large memory overhead.

These state-of-the-art synchrony-based models [25, 24] are not as robust as one might hope. As we show later, these models tend to exploit color as a shortcut feature, a helpful heuristic when different objects have different colors but a harmful one when color is an unreliable shortcut feature. And on more naturalistic datasets, RF largely learns semantic-level groups (see, e.g., Figures 2 & 20 in [25]) rather than instance-level groups which is the central focus of object-centric learning. These state-of-the-art models require either an additional contrastive loss (see, e.g., Figure 8 in [24]) or pre-trained features from self-supervised vision transformers or 'depth masks' even on synthetic datasets (see, e.g., Figure 6 in [25]) to partially resolve such grouping errors.

Many of the complexities introduced in recent models seem to be moving the field away from the core intuition that initially motivated a synchrony-based approach to binding [20]. We step back and describe the original conception of synchrony-based grouping mechanisms from Mozer et al. [20].

Consider the shapes composed of horizontal and vertical bars in Figure 1: the letters $\mathbb{T}$ and $\mathbb{H}$ and a pair of overlapping squares with occlusion. Perceptual grouping involves determining which of the bars belong together. The pairs highlighted in green are part of the same object; the pairs highlighted in red are parts of different objects. A feedforward convolutional architecture is not well suited to this task because spatially local patches are ambiguous with regard to grouping. While a feedforward, fully connected architectures may work in principle, it is also problematic because statistical regularities in images are local and fairly low order (i.e., involve a subset of image features). However, if we drop the feedforward restriction, a convolutional model can iteratively converge on a solution: each iteration suggests soft constraints on how features should be grouped, and over iterations, constraints can propagate from one region in the image to adjacent regions. This methodology was adopted in classical AI vision models [28] and in early synchrony-based models [20]. In the latter, phases—

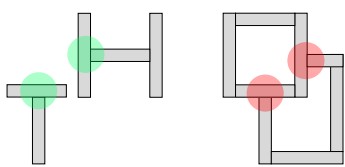

Figure 1: Local feature configurations are insufficient to determine whether features belong to the same object: highlighted horizontal and vertical configurations sometimes belong to the same object (green) and at others to different objects (red).

representing object binding—are initialized to random values and each iteration, the phases are updated to be consistent with constraints from neighboring patches. After multiple iterations the model converges on an interpretation that assigns each image location to a particular phase (object).

Complex weights are a natural choice to instantiate this function in our synchrony formulation outlined above. Just as hidden units in a standard architecture detect configurations of features,

hidden units in a complex-valued analogue detect configurations of features in a particular relative phase arrangement. A hidden unit that is tuned to the top of a $\mathbb{T}$ will expect the horizontal and vertical bar to have the same phase; a hidden unit that is tuned to occlusion (the red patches in Figure 1) will expect the horizontal and vertical bars to be out of phase. In this manner, the hidden units can encode alternative hypotheses concerning object groupings, and iterative processing will attempt to find global configurations that are locally consistent.

Drawing on these intuitions we design a model and training strategy for the fully unsupervised setting. Figure 2 shows our proposed model, **Syn**chronous **C**omple**x** Network (SynCx), which uses complex-valued weights (detectors) to process complex-valued activations (features) in every layer of a fully convolutional autoencoder. The matrix vector product between complex-valued weights and activations ensures that weights process inputs not only based on their features (magnitudes → features) but also their phase relationships (phases → object bindings). The addition and multiplication operators in complex algebra mechanistically implement the binding mechanism in our model removing the need for additional gating mechanisms ($\chi$ [22] / cosine binding [27]) or contrastive training [24]. The spatial locality of constraints are ensured via 2D convolutions used at every layer in the autoencoder. The model is initialized with a random phase map (prior on object bindings) and learns to (de)synchronize groups of features based on their (dis)similarity iteratively. The autoencoder weights are shared across iterations and predicted output phases (top-down feedback) from the previous iteration are used as the input phases at the next iteration while the pixel values are the input magnitudes at each step. Iteratively updating the phases allows spatially local constraints to be slowly propagated across the spatial axes and in principle should lead to the system settling to a fixed point. The autoencoder is trained to simply reconstruct the input image at each iteration.

In contrast to the synchrony formulation of Mozer et al. [20], all recent state-of-the-art models use purely feedforward processing of inputs. Therefore, these models lack the ability to propagate information about local constraints to align phases which we show ($\mathbb{T}$ and $\mathbb{H}$ junction examples) plays a key role in achieving binding via synchrony.

We show that binding in synchrony-based models can be mechanistically implemented simply by matrix-vector products between complex-valued weights and activations to iteratively process inputs. We do not need any additional mechanisms like $\chi$-binding [22, 23, 25] or cosine binding [27] or contrastive training [24] as in prior work to achieve the same. Our conceptually simpler model (SynCx) designed from first principles outperforms (on `Tetrominoes`) or competitive with (on `dSprites` and `CLEVR`) state-of-the-art synchrony-based models for unsupervised object discovery. RF groups objects using largely color cues requiring supervision via additional features ('depth masks') to segregate objects with similar colors. Whereas SynCx more gracefully separates objects of the same color as it re-

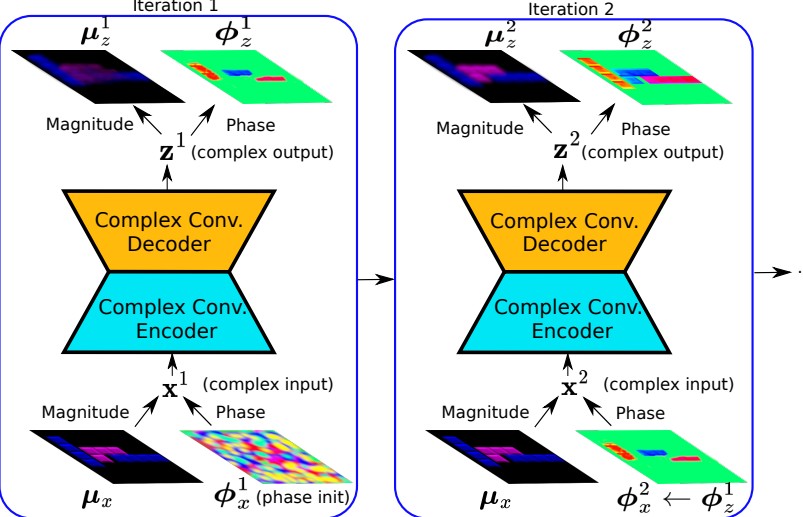

Figure 2: SynCx is a fully convolutional autoencoder that iteratively processes an input image. It starts with a randomly initialized phase $\phi_x^1$ and the input image $\boldsymbol{\mu}_x$ in the magnitude-component updates the phases in a stateful manner, i.e., output phase at iteration 1 fed as input at iteration 2 ($\phi_x^2 \leftarrow \phi_z^1$) and so on. The magnitude-component at the input is always clamped to the input image $\boldsymbol{\mu}_x$. SynCx is trained to reconstruct $\boldsymbol{\mu}_x$ using the output magnitude-component $\boldsymbol{\mu}_z^n$ at every step.

lies on features (color/shape/texture) and the local spatial context removing the need for 'depth masks'. We visualize the phase maps across iterations to qualitatively inspect the binding process. Lastly, we discuss some remaining practical limitations of current state-of-the-art synchrony-based models.

## 2   Method

We describe the intuitions behind our model, its architecture and training procedure below.

**Basic Intuition.**   Our autoencoder model is trained to reconstruct the input image using a representational bottleneck in the hidden layer(s). This requires the model to learn an efficient coding scheme to compress the input image. Images are typically composed of objects which are modular (reusable) units of feature information (shape, color, texture etc.) For a network with complex-valued activations, one intuitive coding scheme is using the phase components to express relationships between lower-order features (edges/textures/shape) in order to compress them into appropriate higher-order features (objects/parts). As the image is processed by the encoder it progressively compresses the input by storing and processing modular 'parts' together as higher-order features while the decoder learns the inverse mapping to recover the input from this compressed format. It is then possible to characterize objects as the collection of features with similar phases across the entire spatial map.

**Model.**   Our model (SynCx) is a fully convolutional autoencoder architecture. It uses *complex-valued* weights to manipulate *complex-valued* activations at each layer. Let $h, w, h', w', c, d_{\text{in}}, d_{\text{out}}$ and $p$ denote positive integers. Every layer of the network is a parametric function $f_{\mathbf{W}} : \mathbb{C}^{h \times w \times d_{\text{in}}} \rightarrow \mathbb{C}^{h' \times w' \times d_{\text{out}}}$ which maps complex-valued inputs $\mathbf{x}$ to complex-valued outputs $\mathbf{h}$ using complex-valued weights $\mathbf{W} \in \mathbb{C}^p$. First, we compute the complex-valued pre-activation response $\mathbf{y}$:

$$\mathbf{y} = f_{\mathbf{W}}(\mathbf{x}), \quad \text{where } f_{\mathbf{W}} \text{ denotes a 2D convolution} \tag{1}$$

Then, we apply the modReLU activation rule [29] element-wise only on the magnitude component of $\mathbf{y}$ (i.e., $\boldsymbol{\mu}_y$) to obtain the complex-valued layer output $\mathbf{h}$ as:

$$\mathbf{h} = \text{modReLU}(\mathbf{y}) = \text{ReLU}(\boldsymbol{\mu}_y + b) \odot e^{i\boldsymbol{\phi}_y} \;\; \text{with} \;\; \mathbf{y} \equiv \boldsymbol{\mu}_y \odot e^{i\boldsymbol{\phi}_y} \tag{2}$$

where $\odot$ denotes a Hadamard product and $b \in \mathbb{R}^{d_{\text{out}}}$ is a learnable parameter. We use modReLU instead of ReLU because the nonlinearity is applied only to the magnitude component (strictly non-negative). Consequently, the standard ReLU activation rule would operate only in its linear region. The modReLU activation rule [29] was introduced specifically for a setting such as ours where the activation function is applied only to the magnitude components of complex-valued activations. The encoder module progressively downsamples the resolution of the complex-valued spatial feature map using strided 2D convolutions. The decoder module upsamples the resolution of the spatial feature map using the upsampling function independently on its magnitude and phase components.

**Training and Loss Function.**   Given an image $\boldsymbol{\mu}_x \in \mathbb{R}^{h \times w \times c}$ of height $h$, width $w$, and $c$ channels (3 for color images) with non-negative pixel values. We construct a complex-valued input with magnitudes $\boldsymbol{\mu}_x$ and phases $\boldsymbol{\phi}_x$ which are randomly initialized at every spatial location and channel of the image (see Figure 2). For each vector-valued feature at a given location, its phase component encodes the network's current hypothesis about its object binding; random initialization reflects lack of knowledge initially. The autoencoder processes the input image to update its hypotheses about object bindings at every iteration $n \in \{1, ..., N\}$. The complex-valued input $\mathbf{x}^n \in \mathbb{C}^{h \times w \times c}$ to the autoencoder at the $n^{\text{th}}$ iteration is:

$$\mathbf{x}^n = \boldsymbol{\mu}_x \odot e^{i\boldsymbol{\phi}_x^n} \tag{3}$$

Notice in Equation (3), that the magnitude components are always clamped to the image ($\boldsymbol{\mu}_x$) while the phase components ($\boldsymbol{\phi}_x^n$) are fed back. At the $n^{\text{th}}$ iteration, $\mathbf{x}^n$ is processed by the encoder and decoder layers, denoted as $\text{Net}(\mathbf{x}^n)$, to compute a complex-valued output $\mathbf{z}^n \in \mathbb{C}^{h \times w \times c}$. The magnitude component of $\mathbf{z}^n$, i.e. $\boldsymbol{\mu}_z^n$, is the reconstruction of input image $\boldsymbol{\mu}_x$. The phase component of $\mathbf{z}^n$, i.e. $\boldsymbol{\phi}_z^n$, is used to initialize the complex-valued input $\mathbf{x}^{n+1}$ at the next iteration:

$$\mathbf{z}^n = \text{Net}(\mathbf{x}^n) \quad ; \quad \boldsymbol{\phi}_x^{n+1} \leftarrow \boldsymbol{\phi}_z^n \tag{4}$$

This way the phase maps act as the state of a recurrent function which processes a clamped input image at all steps as shown in Figure 2. After $N$ such iterative updates to the phase maps by the

autoencoder, it is trained to minimize the average pixel-wise reconstruction loss across all iterations:

$$\mathcal{L} = \frac{1}{N} \sum_{n=1}^{N} ||\boldsymbol{\mu}_x - \boldsymbol{\mu}_z^n||^2 \tag{5}$$

We use the weight initialization scheme outlined by Trabelsi et al. [30] for the complex-valued weights $\mathbf{W}$ at every layer of the encoder and decoder modules. This initialization scheme derives the variances of complex-valued weights to satisfy the initialization criterion of He et al. [31]. The magnitude components of the complex-valued weights at every layer are sampled from a Rayleigh distribution with $\sigma = 1/\text{fan}_{\text{in}}$ while the phase components are sampled from a uniform distribution between 0 to $2\pi$.

Through ablation studies in the following section, we quantify the effects of key components such as random initialization of the phase map, presence of a bottleneck at latent layer(s) and the number of iterative updates have on the performance of our system.

**Extracting Object Assignments.** Here, we outline the processing steps used to extract discrete object assignments for every pixel given its continuous phase values. We start by taking the output complex-valued feature map $\mathbf{h}^N \in \mathbb{C}^{h \times w \times d_{\text{out}}}$ from the penultimate decoder layer at the last iteration $N$. We use latent-level complex-valued features instead of output-level so as to use high-order features (texture/shape/color etc.) than simply color cues (RGB space) to determine their object constituency. Then, we construct a complex-valued feature map with unit magnitudes (i.e., $\boldsymbol{\mu}_h^N \leftarrow \mathbf{1} \in \mathbb{R}^{h \times w \times d_{\text{out}}}$) and phases the same as that of $\mathbf{h}^N$ (i.e., $\boldsymbol{\phi}_h^N$). Background regions in this constructed feature map are masked out by setting their magnitudes to zero and then converted to the Cartesian form element-wise. Finally, these features in Cartesian form are clustered using $k$-Means to compute the object assignments for every location in the spatial map. Using unit magnitudes for complex-valued activations ensures that object assignments are solely determined by their orientations. We note that it would be most natural to cluster the phases from the output layer at each location to assign it to an object. However, we observed that the object separation shown in the output phases is weaker than that from the high-dimensional latent layer. Therefore, for all results presented here we choose to cluster the higher dimensional latent representations (from the penultimate decoder layer) by location. For more details on the object assignment process refer to Appendix A.

## 3 Results

First, we describe details of the datasets, our model/baselines, training and evaluation procedures. We visualize the phase maps and qualitatively characterize the grouping behavior shown by our model. We quantitatively compare the grouping performance on the unsupervised object discovery task of our model (SynCx) against state-of-the-art synchrony-based baselines (CAE, CAE++, CtCAE, and RF). We also quantify the effects of key components of our model using ablation experiments. We conclude with some practical limitations shown by the binding mechanisms in these synchrony-based models.

**Datasets.** We evaluate models on three datasets from the multi-object suite [32] namely `Tetrominoes`, `dSprites` and `CLEVR` used by prior work in object-centric learning [33–35]. For `CLEVR`, we use a filtered version [35] which consists of images containing less than seven objects. In all experiments we use image resolutions identical to Emami et al. [35], i.e., 35x35 for `Tetrominoes`, 64x64 for `dSprites` and 96x96 for `CLEVR` (center crop of 192x192 resized to 96x96). In `Tetrominoes` and `dSprites` the number of training images is 60K whereas in `CLEVR` it is 50K. All three datasets have 320 test images on which we report the evaluation metrics. For further details about datasets and preprocessing, we refer to Appendix A.

**Model & Training Details.** For more details about the encoder and decoder architecture of our model we refer to Appendix A. We train our model for 40K steps on `Tetrominoes`, and 100K steps on `dSprites` and `CLEVR` with Adam optimizer [36] with a constant learning rate of 5e-4, i.e., no warmup schedules or decay (all hyperparameter details are given in Appendix A). The phase initialization at each spatial location and channel of the input image are independent samples from a von-Mises distribution with a mean of 0 and concentration of 1.

**Evaluation Metrics.** We use the same evaluation protocol as prior work [33–35, 23] which compares the grouping performance of models using the Adjusted Rand Index (ARI) [37, 38]. The ARI scores are measured only for the foreground pixels common practice in the object-centric literature.

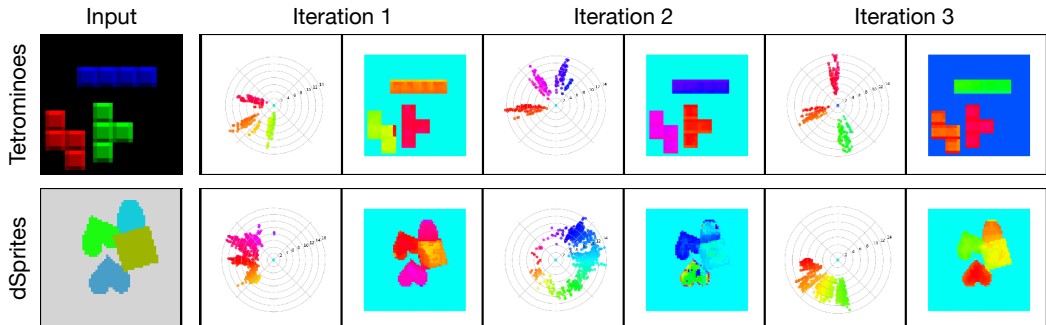

Figure 3: Evolution of phase maps in radial and heatmap form (colors matched) across iterations in SynCx for two inputs from `Tetrominoes` (row 1) and `dSprites` (row 2).

**Phase Map Visualization.** We seek to qualitatively inspect phase components of the output complex-valued feature map $\mathbf{h} \in \mathbb{C}^{h' \times w' \times d_{out}}$ from an intermediate decoder layer to see if our model has indeed learned phases specialized in an object-centric manner. We visualize the phase component of the high-dimensional feature map as a heatmap in two dimensions. We assign unit magnitudes and retain the phase components of $\mathbf{h}$ and elementwise convert these to the cartesian form as we did previously to extract object assignments. We apply t-SNE [39] to perform dimensionality reduction on the complex-valued feature map of an input image to recover a scalar 'composite' phase value at every spatial location (see Appendix A and Figure 10 for more details). This 'composite' phase map is visualized as a two dimensional heatmap and radial plot with colors in the former corresponding to orientations in the latter (see columns 2 & 3 in Figure 3). From Figure 3 shows how the phase maps evolve across the iterations in SynCx grouping process. In the sample image from `Tetrominoes` (row 1 in Figure 3), we can see how the phase 'bands' corresponding to each tetris block are progressively separated in their orientations. Such a visualization gives an interpretable artefact that allows a qualitative inspection of the phase synchronization process.

**Unsupervised Object Discovery.** Table 1 compares the performance of recent state-of-the-art synchrony-based baselines (CAE, CAE++, CtCAE, and RF) against ours (SynCx) on the unsupervised object discovery task. On `Tetrominoes`, we observe that SynCx outperforms all baselines on grouping performance. SynCx more gracefully separates objects of the same color (see Figure 4) compared to RF which in addition to complex-valued activations requires $\chi$-binding mechanism. SynCx also outperforms the CtCAE baseline which in addition to complex-valued activations requires $\chi$-binding and contrastive training to separate similarly colored objects. On `dSprites`, SynCx significantly outperforms CAE, CAE++ and CtCAE baselines while being strongly competitive with RF. Similarly on `CLEVR`, SynCx strongly outperforms CAE, CAE++ and CtCAE baselines while being competitive with RF. Overall, SynCx despite its simple binding mechanism (no gates) and training strategy (no contrastive training) outperforms or is strongly competitive with all recent state-of-the-art synchrony-based models. However, there still remains a significant gap in grouping performance between synchrony-based models w.r.t dominant slot-based approaches such as SlotAttention [34] especially on `CLEVR`. We refer to Appendix B for the model comparisons that includes the MSE loss as well. We refer to Appendix C for additional qualitative grouping examples from our model.

Table 1: ARI scores (mean $\pm$ standard deviation across 5 seeds) for CAE, CAE++, CtCAE, RF, SynCx and SlotAttention on `Tetrominoes`, `dSprites` and `CLEVR`. Results for CAE, CAE++ and CtCAE baselines taken from Stanić et al. [24] and for SlotAttention taken from Locatello et al. [34].

| Model | Tetrominoes | dSprites | CLEVR |
|---|---|---|---|
| CAE | $0.00 \pm 0.00$ | $0.05 \pm 0.02$ | $0.04 \pm 0.03$ |
| CAE++ | $0.78 \pm 0.07$ | $0.51 \pm 0.08$ | $0.27 \pm 0.13$ |
| CtCAE | $0.84 \pm 0.09$ | $0.56 \pm 0.11$ | $0.54 \pm 0.02$ |
| RF | $0.42 \pm 0.09$ | $0.84 \pm 0.03$ | $0.65 \pm 0.01$ |
| SynCx | $0.89 \pm 0.01$ | $0.82 \pm 0.01$ | $0.59 \pm 0.03$ |
| SlotAttention | $0.99 \pm 0.01$ | $0.91 \pm 0.01$ | $0.99 \pm 0.01$ |

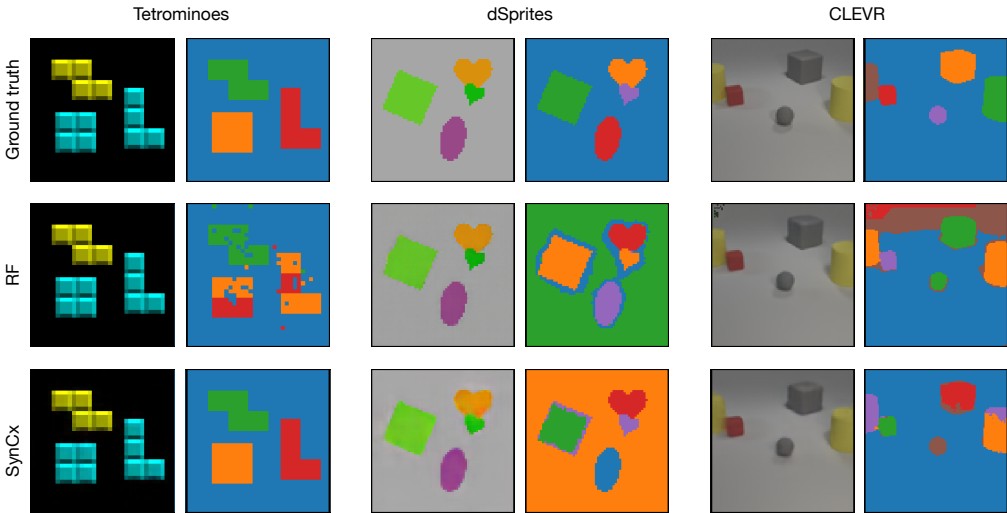

Figure 4: Comparison between RF and SynCx grouping on `Tetrominoes`, `dSprites` and CLEVR. RF tends to systematically group similarly colored objects together while SynCx is more adept at separating them such as blue tetris blocks (left), green square and heart (middle) and yellow cylinders (right).

Below, we quantify the effects of key components of our model such as representational bottlenecks, number of iterations and phase initialization have on its grouping performance.

**Effect of Bottlenecks.** We measure the effect of having representational bottlenecks (i.e. spatial resolution of feature maps) in the hidden layers of our model. To do so, we design a variant of our model (denoted as SynCx w/o bottleneck in Table 2) that preserves the spatial resolution of feature maps in the encoder (using stride of 1 in all layers) thereby removing all bottlenecks.

We ensure that both these variants (SynCx and SynCx w/o bottleneck) have the same number of parameters. Instead we only vary the spatial resolution of feature maps, so the bottleneck is w.r.t spatial resolution and not the number of model parameters. We observe a sharp drop in grouping performance for our ablated

Table 2: Bottleneck ablation on `Tetrominoes`.

| Model | MSE $\downarrow$ | ARI $\uparrow$ |
|---|---|---|
| SynCx w/o bottleneck | 6.29e-5 $\pm$ 9.00e-5 | 0.10 $\pm$ 0.06 |
| SynCx | 2.07e-3 $\pm$ 1.09e-4 | 0.89 $\pm$ 0.01 |

model variant without any bottlenecks despite it achieving a lower test loss (see Figure 5). This result supports our initial intuition that representational bottlenecks force an autoencoder with complex-valued activations to compress spatial regularities in features by using phase components to capture relationships between them. Since objects are modular units containing highly regular features within their boundaries, this leads to phases strongly specializing (synchronizing) towards them to facilitate better image compression.

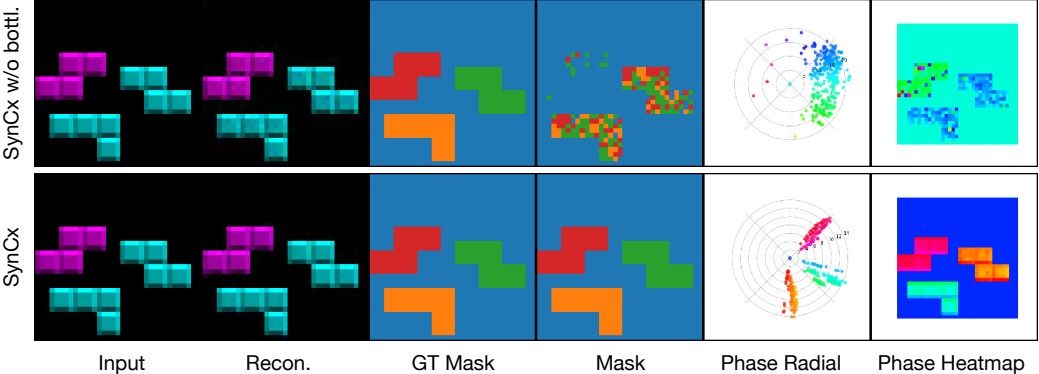

Figure 5: Reconstruction, object masks, radial phase plot and phase heatmaps (colors matched between columns 5 & 6) for SynCx without the bottleneck (row 1) and the full SynCx model (row 2).

**Effect of Iterations.** We measure the effect of multiple iterations to update phases on the grouping performance of our model. Table 3 shows the grouping performance of our model with increasing number of iterations used during training. We observe a general trend of increasing phase specialization towards objects and better reconstructions as we increase the number of iterations. This supports our qualitative characterization of the models' operation — starting with random object assignments it iteratively refines the hypotheses for the assignment for each pixel.

Table 3: Ablation on number of iterations used for training on `dSprites`.

| No. of Iterations | MSE $\downarrow$ | ARI $\uparrow$ |
|:---:|:---:|:---:|
| 1 | 3.49e-3 $\pm$ 1.11e-4 | 0.57 $\pm$ 0.01 |
| 2 | 2.20e-3 $\pm$ 6.24e-5 | 0.81 $\pm$ 0.03 |
| 3 | 1.73e-3 $\pm$ 1.42e-4 | 0.82 $\pm$ 0.01 |
| 4 | 1.73e-3 $\pm$ 3.85e-4 | 0.79 $\pm$ 0.03 |

We also measure the effect of increasing the number of iterations at test-time compared to that used during training. Table 4 shows the grouping performance of our model as we increase the number of iterations from $4$ to $6$ while during training the model used 3 iterations. We observe no drop in object separation performance of our model when we extrapolate the number of iterations used at test-time.

Table 4: Ablation on number of iterations used at test-time on `dSprites`.

| No. of Iterations | MSE $\downarrow$ | ARI $\uparrow$ |
|:---:|:---:|:---:|
| 4 | 1.71e-3 $\pm$ 1.38e-4 | 0.81 $\pm$ 0.01 |
| 5 | 1.71e-3 $\pm$ 1.40e-4 | 0.82 $\pm$ 0.01 |
| 6 | 1.70e-3 $\pm$ 1.41e-4 | 0.82 $\pm$ 0.01 |

**Effect of Phase Initialization.** We measure the effect that the initialization of phases has on the grouping performance of our model. To achieve this, we compare (see Table 5) the grouping performance of three variants — i) initial phases sampled from a uniform distribution between -$\pi$ and $\pi$ ii) initial phases sampled from a von-Mises distribution with mean of $0$ and concentration of $1$ and iii) all values in the initial phase map are set to zero.

Rest of the model and training hyperparameters are kept the same between these two variants. We observe that the variant using von-Mises distribution to sample initial input phases achieves the best test loss and improved grouping scores among all variants. This suggests the variance of the noise distribution is an important factor to tune for phase synchronization. The von-Mises

Table 5: Phase init. ablation on `Tetrominoes`.

| Phase Init. | MSE $\downarrow$ | ARI $\uparrow$ |
|:---:|:---:|:---:|
| Zero | 2.46e-3 $\pm$ 5.08e-7 | 0.74 $\pm$ 0.04 |
| Uniform | 1.02e-2 $\pm$ 8.07e-3 | 0.75 $\pm$ 0.05 |
| von-Mises | 2.07e-3 $\pm$ 1.09e-4 | 0.89 $\pm$ 0.01 |

distribution with a mean of $0$ and concentration of $1$ is the circular analogue of the Normal distribution and therefore samples "noisy" phase values with an intermediate level of variance compared to the other two alternatives which have maximum and zero variance. We use the von-Mises variant as the default phase initialization for input phases in all experiments.

Table 6: Comparison of parameter counts (rounded up to the nearest thousand) of various synchrony-based models. Total number of parameters expressed in terms of number of real-valued floats.

| Model | Tetrominoes | dSprites | CLEVR |
|:---:|:---:|:---:|:---:|
| CAE++ | 5,372,000 | 11,713,000 | 22,281,000 |
| CtCAE | 5,372,000 | 11,713,000 | 22,281,000 |
| RF | 6,630,000 | 11,861,000 | 22,592,000 |
| SynCx | 966,000 | 974,000 | 974,000 |

**Parameter and Training Efficiency** Table 6 shows the parameters counts for the various synchrony-based models. We see that our model, SynCx, is 6-23x more parameter efficient compared to the other synchrony-based alternatives. Table 7 shows the wall-clock training times for convergence of SynCx against the strongest synchrony baseline, RF, on `Tetrominoes` and `dSprites`. We see that our model on these datasets consistently takes lesser wall-clock time for training in these settings.

Table 7: Wall-clock training times for SynCx and RF models on a P100 GPU.

| Model | Tetrominoes | dSprites |
|:---:|:---:|:---:|
| RF | 7h 20m | 4h 50m |
| SynCx | 1h 35m | 4h 10m |

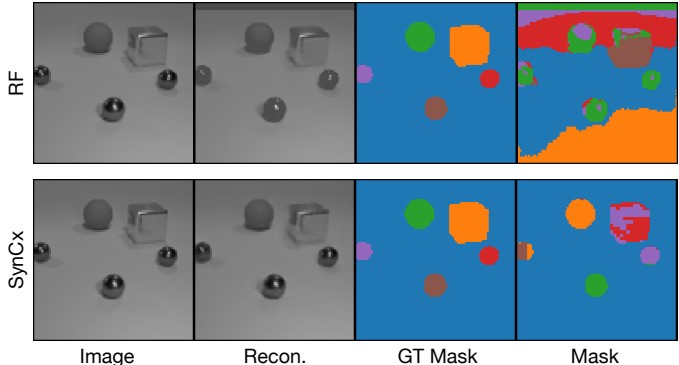

Figure 6: Reconstruction and object masks for RF and SynCx on grayscale `CLEVR`. While both models reconstruct well, RF struggles to group the objects without access to color unlike SynCx.

These results highlight the computational benefits of the simple ideas of complex-valued weights and recurrence that are the essence of our model design.

**Limitations.** We test how reliant on color cues are the two best performing synchrony-based models from Table 1, i.e., RF and SynCx, for unsupervised object discovery. We train and evaluate them on a grayscale version of the `CLEVR` dataset. To separate objects in this dataset, the models' would need to rely more on other features like shape, texture, etc. From Table 8, we see that the grouping performance of RF on grayscale `CLEVR` drops sharply compared to the original (see Table 1) indicating its heavy reliance on color cues. Our models' grouping performance also drops but to a much lesser degree compared to RF. This indicates a weaker dependence on color cues in the binding mechanism of our model. In fact, on grayscale `CLEVR` SynCx significantly outperforms RF despite having slightly worse performance on the original colored version.

The binding mechanism (i.e., matrix-vector product) in our model updates phases based on agreement between feature detectors (complex-valued weights) and high-dimensional attributes (complex-valued activations) that model color, edges, texture, shape etc. while taking it account their phase relationships (local context). This facilitates it to bind to objects more robustly and not simply rely on color. Overall, we can see that despite its simple architecture and

Table 8: Grayscale `CLEVR`.

| Model | MSE $\downarrow$ | ARI $\uparrow$ |
|---|---|---|
| RF | 5.64e-4 $\pm$ 3.76e-4 | 0.22 $\pm$ 0.04 |
| SynCx | 2.49e-4 $\pm$ 3.76e-5 | 0.45 $\pm$ 0.01 |

training procedure our model shows good grouping performance on a wider range of visual environments. However, binding mechanisms in state-of-the-art synchrony models are far from perfect. They are unable to reliably capture objects even on synthetic datasets like `CLEVR` containing simple 3D shapes with metallic or matte textures under camera lighting with a non-textured background.

## 4 Related Work

**Slot-based Binding.** Recent years have seen a growing number of models for unsupervised perceptual grouping (see Greff et al. [10] for a summary). These models maintain the separation of information using a separate set of activations ('slots') but differ in the segregation mechanism to infer their contents. Representational symmetry in 'slots' is broken via temporal ordering [40, 41], spatial ordering [42–44], type-specificity [45], iterative routing procedures [46, 47, 33, 34, 48] (cf. earlier work [49]) or combinations thereof [50–52]. Extensions of slot-based grouping models for videos [53–56], novel view synthesis [57–59] as well as other modalities than vision including speech [60], music [61] and actions [62] have been explored. Conceptual limitations of slot-based models include separation only maintained at one level, binding information stored in hard-wired architectural components unsuited for gradient-based adaptation, uniform capacity, inability to store object-level relational factors and high computational cost for training as noted by Stanić et al. [24].

**Synchrony-based Binding.** Synchrony-based models could in principle address the above shortcomings of slot-based models but have so far received comparatively little attention. They resolve the binding problem by augmenting each activation with additional grouping features. They can

be broadly divided into two classes — *temporal* and *complex*, based on the type of coding strategy used for the augmentation. Models using temporal codes rely on spiking neurons with rhythmic firing behavior [63–65]. In such models, features of an object are expressed by neurons that fire in-sync. However, spiking neurons are non-differentiable and therefore incompatible with gradient-based learning, requiring specialized learning algorithms difficult to scale with current hardware accelerators for training. Models using complex-valued codes [20, 21, 66, 22, 23] are based on complex-valued activations suitable for scalable training using backpropagation. These models have been empirically benchmarked only on binarized images of simple geometric shapes or MNIST digits. More recent models [25, 24] show improved unsupervised grouping performance on more visually challenging color images from the multi-object suite [32]. However, these recent models heavily rely on supervision in the form of 'depth masks', gating mechanisms, contrastive training or a combination thereof. In contrast, our model is simply a fully convolutional autoencoder with complex-valued weights that iteratively updates phases to reconstruct an input image.

**Temporal Correlation Hypothesis.** Synchrony-based models draw functional inspiration from the *temporal correlation hypothesis* [67, 68] in neuroscience. It posits that the brain uses synchronized dynamics of neuronal firings to bind together distributed feature information computed in parallel at different areas into coherent percepts. Neural synchrony is also believed to convey information about relationships between features needed for dynamic and context-dependent binding by *'relational coding'* [69]. This is in contrast to *'labeled line coding'* where each unit has a fixed label attached to it indicating the static feature conjuction being active. Our use of complex-valued weights to process complex-valued activations is akin to the *'relational coding'* scheme.

## 5    Conclusion

We explored synchrony-based binding with an architecture, SynCx, that differs from current models in three respects: i) SynCx's weights are complex valued, allowing it to encode joint feature-phase configurations in the weights; ii) SynCx is recurrent and stateful, allowing it to perform iterative constraint propagation; and iii) SynCx has an internal representational bottleneck, which require it to use the phases of complex-valued activations to encode statistical regularities in images. These three properties are crucial to extending object-centric phase synchronization behavior to the fully unsupervised setting. Our conceptually elegant model outperforms all state-of-the-art baselines on `Tetrominoes` and overcomes a common failure mode of these baselines in which similarly colored objects are grouped together. Overreliance on color cues for grouping by current models is further highlighted on a grayscale variant of the `CLEVR` dataset where we see that the RF model struggles to group objects using other features (shape and texture), in contrast to SynCx. Our model shows strong performance compared to more sophisticated synchrony-based baselines [25, 24] on `dSprites` and `CLEVR` without the need for additional supervision such as 'depth masks', gating mechanisms or contrastive training. Starting with our model as a simple template, future directions to explore include the use of a temporal difference style loss to weight reconstruction targets and incorporate spatial priors into the binding mechanism. The process of extracting discrete object assignments from continuous phase maps by clustering could be improved by accounting for outliers, non-Gaussian distributed phase values with unequal sizes and variances. While recent synchrony-based models have been steadily closing the gap in grouping performance to well-established slot-based approaches such as SlotAttention [34] on simple synthetic datasets (`Tetrominoes` and `dSprites`) the gap remains significant on `CLEVR`. Further, they are yet to be scaled up to more challenging multi-object visual benchmarks (e.g. MOVi [70]) which offers an open challenge for future work.

**Acknowledgments.** We thank Kazuki Irie for numerous discussions and guidance. We also thank Thomas Kipf for giving valuable feedback on our manuscript. This research was funded by Swiss National Science Foundation grant: 200021_192356, project NEUSYM and partially by the ERC Advanced grant no: 742870, AlgoRNN. This work was also supported by the following grants from the Swiss National Supercomputing Centre (CSCS) under project ID s1205 and s1294. We also thank NVIDIA Corporation for donating DGX machines as part of the Pioneers of AI Research Award.

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

# A   Experimental Details

**Datasets.**   We evaluate all models on a subset of three datasets from the multi-object suite [32] namely: `Tetrominoes` which consists of colored tetris blocks on a black background, `dSprites` with colored sprites of various shapes like heart, square, oval, etc., on a grayscale background, and lastly, `CLEVR`, a dataset from a synthetic 3D environment. For `CLEVR`, we use the filtered version [35] which consists of images containing less than seven objects sometimes referred to as `CLEVR6` as in Locatello et al. [34]. We normalize all input RGB images to have pixel values in the range $[0, 1]$ consistent with prior work [23]. To generate the grayscale variant of the `CLEVR` dataset we apply the color to grayscale conversion function from the `Pillow` library as part of the data preprocessing pipeline.

**Models.**   Table 9 shows the architecture specifications such as number of layers, kernel sizes, stride lengths, number of channels etc., for the convolution layers used by the encoder and decoder modules in SynCx. We reproduce unsupervised object discovery results of the RF baseline by adapting the source code released by the authors. [4] Similarly, we reproduce unsupervised object discovery for the CAE, CAE++ and CtCAE by adapting the source code released by the authors. [5]

Table 9: Encoder and Decoder architecture specifications for SynCx.

| **Encoder** |
| --- |
| $3 \times 3$ conv, 64 channels, stride 2, modReLU |
| $3 \times 3$ conv, 128 channels, stride 2, modReLU |
| $3 \times 3$ conv, 128 channels, stride 2, modReLU |
| **Decoder** |
| Nearest neighbor upsample x2 |
| $3 \times 3$ conv, 128 channels, stride 1, modReLU |
| Nearest neighbor upsample x2 |
| $3 \times 3$ conv, 64 channels, stride 1, modReLU |
| Nearest neighbor upsample x2 |
| $3 \times 3$ conv, 64 channels, stride 1, modReLU |
| *For 64×64 and 96×96 inputs, 1 additional decoder layer:* 
 $1 \times 1$ conv, 64 channels, stride 1, modReLU |
| **Output Layer** |
| $1 \times 1$ conv, 3 channels, stride 1 |

**Training Details.**   Table 10 shows the hyperparameter configurations used to report unsupervised object discovery results (Table 1) for SynCx. To reproduce the unsupervised object discovery results for RF in Table 1 we use the hyperparameter configurations reported by the authors [25] listed in Appendix D.6.

Table 10: Training hyperparameters for SynCx.

| Hyperparameter | Tetrominoes | dSprites | CLEVR |
| --- | --- | --- | --- |
| Training Steps | 40,000 | 100,000 | 100,000 |
| Batch size | 64 | 16 | 32 |
| Learning rate | 5e-4 | 5e-4 | 5e-4 |
| Gradient Norm Clipping | 1.0 | 1.0 | 1.0 |
| Number of iterations | 3 | 3 | 4 |
| Phase initialization | von-Mises | von-Mises | von-Mises |

---

[4] `https://github.com/loeweX/RotatingFeatures`
[5] `https://github.com/agopal42/ctcae`

**Computational Efficiency.**   We report the training and inference time (wall-clock) for our models across 3 image resolutions for the 3 datasets used in this work from the multi-object suite. Inference on the test set containing 320 images of 35x35 resolution takes 7.87 seconds using 3 iterations, for 64x64 images it takes 38.96 seconds using 3 iterations and for the 96x96 images it takes 112.75 seconds using 4 iterations on a single NVIDIA GTX1080Ti GPU. Training time(s) on the other hand differs depending on the image resolution, number of iterations and model size. To train our model for 40k steps on 35x35 resolution images from `Tetrominoes` took 1.7 hours on a NVIDIA Tesla P100 GPU. To train our model for 100k steps on 64x64 resolution images from `dSprites` took 4.25 hours on a NVIDIA Tesla P100 GPU. To train our model for 100k steps on 96x96 resolution images from `CLEVR` took 17.87 hours on a NVIDIA Tesla V100-SXM2 GPU. To reproduce all the results/tables (mean and std-dev across 5 seeds) reported in this work we estimate the compute requirement to be 390 GPU hours in total for training models (RF and SynCx). Further, we estimate that the total compute used in this project is roughly 10-15 times more than the above figure, used for experiments in the prototyping phase. Note, that the figure above does not account for the phase map visualization and evaluation (clustering) procedures to extract object masks using the trained models primarily executed on CPUs.

**Phase Map Visualization.**   We seek to visualize the complex-valued feature map $\mathbf{h} \in \mathbb{C}^{h' \times w' \times d_{\text{out}}}$ at some intermediate layer of the decoder module as an image. We apply dimensionality reduction along the channels of $\mathbf{h}$ so as to represent it as an one dimensional feature map. We start by constructing a complex-valued feature map in polar form with phase components $(\phi_h)$ and magnitude components being the identity, i.e., $\boldsymbol{\mu}_h \leftarrow \mathbf{1} \in \mathbb{R}^{h' \times w' \times d_{\text{out}}}$. Then, we convert the feature map obtained in the previous step from polar to Cartesian form. Next, we apply t-SNE to project the Cartesian domain feature map in two dimensions at each spatial location. We use the t-SNE implementation in the `scikit-learn` library [71] (`sklearn.manifold.TSNE`) with `n_iter` set to 500, `metric` set to Euclidean and `perplexity` set to 20. We also experimented with UMAP [72] for the dimensionality reduction computation but found it to be inferior to t-SNE qualitatively (see Figure 10) and processing time per image (UMAP takes $\approx 24$ seconds whereas t-SNE takes $\approx 8$ seconds). We use the UMAP implementation in the `umap-learn` library [73] (`umap.UMAP`) with default arguments. Therefore, we use t-SNE for dimensionality reduction in the phase visualization process. We recover 'composite' phase $\overline{\boldsymbol{\phi}}_h \in \mathbb{R}^{h' \times w'}$ by converting the two dimensional feature back to the polar form. The 'composite' magnitude $\overline{\boldsymbol{\mu}}_h \in \mathbb{R}^{h' \times w'}$ at each spatial location of the map is the Euclidean norm (along channels) of the vector-valued features (i.e., magnitude components of $\mathbf{h}$) at that location. The 'composite' phase at every spatial location is visualized as the heatmap value and the 'composite' magnitude is its distance from the center in the radial plot. The colors of points are matched across the heatmap and the radial plot such that the color of a pixel in the former correspond to its orientation in the latter. Lastly, the magnitude of background regions are masked out as was the case while extracting object assignments from phase maps.

**Evaluation Details.**   To evaluate our model for the unsupervised object discovery task we require discrete object assignments at every location in the image. We cluster the continuous phase components of the complex-valued feature maps from an intermediate layer (second to last by default) of the decoder to compute these object assignments for every pixel. We use the $k$-means implementation in the scikit-learn library [71] (`sklearn.clustering.KMeans`) with `n_clusters` set using the ground-truth value for each dataset and `n_init` set to 5.

# B   Additional Results

Table 11: MSE and ARI scores (mean $\pm$ standard deviation across 5 seeds) for CAE, CAE++, CtCAE, RF and SynCx models for Tetrominoes, dSprites and CLEVR. Results for CAE, CAE++ and CtCAE baselines are taken from Stanić et al. [24].

| Dataset | Model | MSE ↓ | ARI ↑ |
|---|---|---|---|
| Tetrominoes | CAE | 4.57e-2 $\pm$ 1.08e-3 | 0.00 $\pm$ 0.00 |
| | CAE++ | 5.07e-5 $\pm$ 2.80e-5 | 0.78 $\pm$ 0.07 |
| | CtCAE | 9.73e-5 $\pm$ 4.64e-5 | 0.84 $\pm$ 0.09 |
| | RF | 5.27e-6 $\pm$ 2.60e-6 | 0.42 $\pm$ 0.09 |
| | SynCx | 2.07e-3 $\pm$ 1.09e-4 | 0.89 $\pm$ 0.01 |
| dSprites | CAE | 8.16e-3 $\pm$ 2.54e-5 | 0.05 $\pm$ 0.02 |
| | CAE++ | 1.60e-3 $\pm$ 1.33e-3 | 0.51 $\pm$ 0.08 |
| | CtCAE | 1.56e-3 $\pm$ 1.58e-4 | 0.56 $\pm$ 0.11 |
| | RF | 5.96e-4 $\pm$ 1.62e-4 | 0.84 $\pm$ 0.03 |
| | SynCx | 1.73e-3 $\pm$ 1.42e-4 | 0.82 $\pm$ 0.01 |
| CLEVR | CAE | 1.50e-3 $\pm$ 4.53e-4 | 0.04 $\pm$ 0.03 |
| | CAE++ | 2.41e-4 $\pm$ 3.45e-5 | 0.27 $\pm$ 0.13 |
| | CtCAE | 3.39e-4 $\pm$ 3.65e-5 | 0.54 $\pm$ 0.02 |
| | RF | 2.69e-4 $\pm$ 9.37e-5 | 0.65 $\pm$ 0.01 |
| | SynCx | 3.30e-4 $\pm$ 5.30e-5 | 0.59 $\pm$ 0.03 |

# C  Additional Visualizations

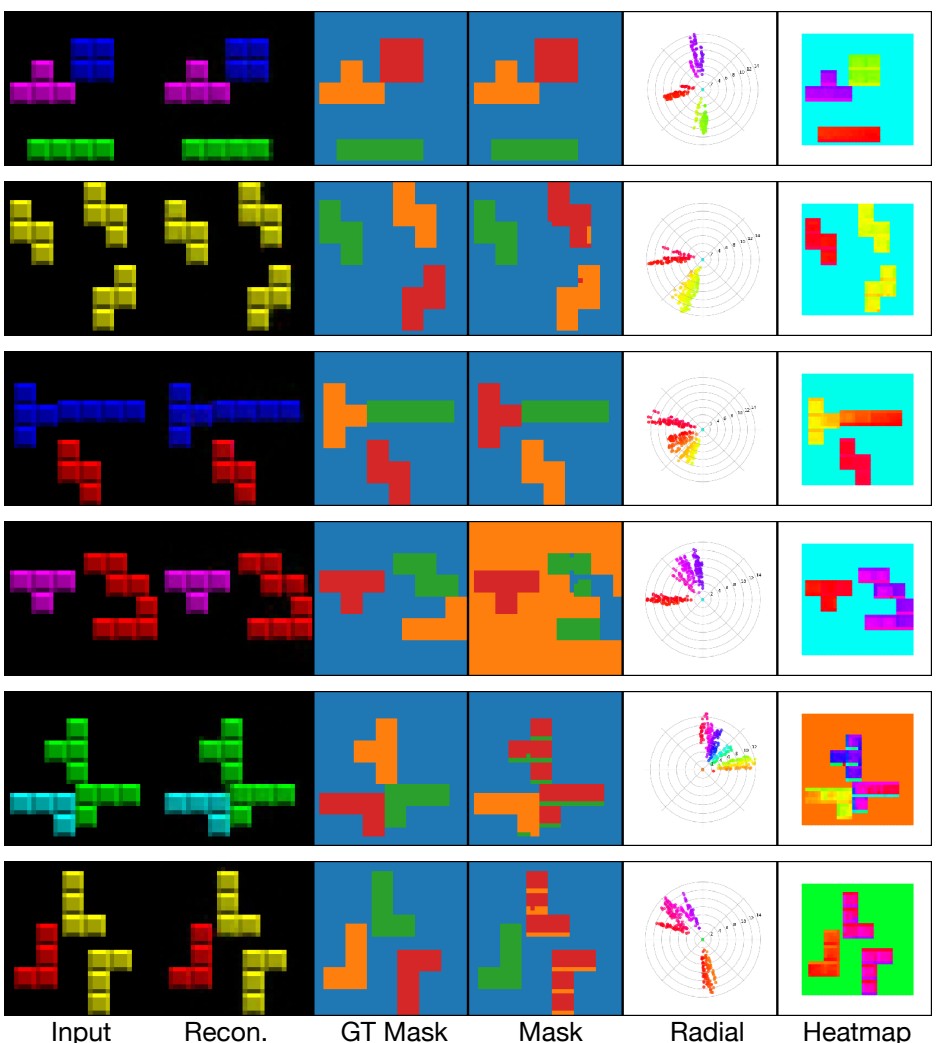

Figure 7: Collection of images grouped by SynCx from the `Tetrominoes` dataset. Rows 1-3 show cases where SynCx perfectly groups the 3 tetris blocks even with mulitple objects of the same color. Row 4 shows a failure mode where it imperfectly partitions the two red blocks into one straight parts and two $\mathbb{L}$-shaped parts. It is plausible decomposition since the dataset contains many tetris blocks with such $\mathbb{L}$-shaped bends. Rows 5 and 6 show another failure mode where it fails to decompose the two similarly colored tetris blocks at all. Rather SynCx learns specialized phases for the edges and interior portions of each of the similarly colored tetris blocks. This could be a caused by a particularly poor initialization of phase maps by random sampling.

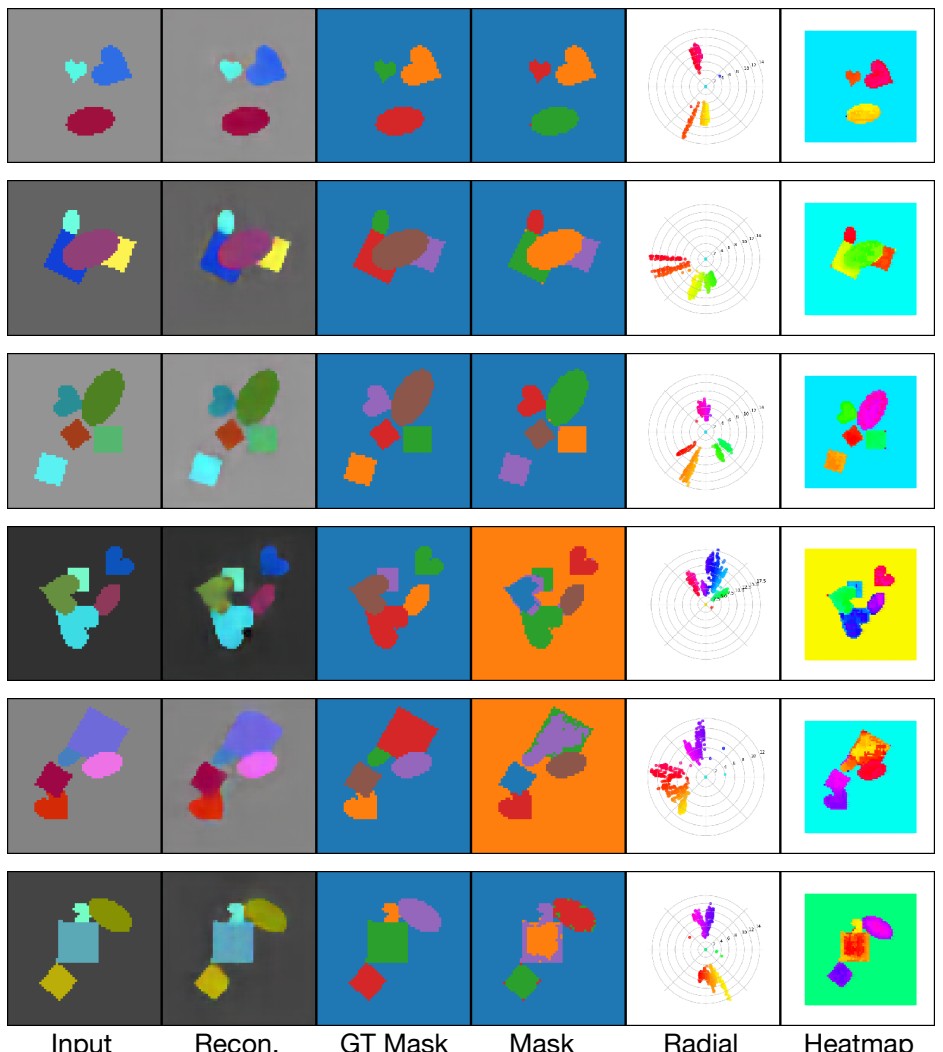

| Input | Recon. | GT Mask | Mask | Radial | Heatmap |

Figure 8: Collection of images grouped by SynCx from the `dSprites` dataset. Rows 1-3 show cases where SynCx perfectly groups images containing 3-5 objects. We can also see that the pairs of phase map and radial plot visualizations show discernible specialization in phase values towards an object. Row 4 shows a grouping failure where the square and heart in light blue are incorrectly grouped together. However, on closer inspection of the corresponding phase plots we can see that the phases do show specialization (see row 4 column 6) towards both the square (light blue) and heart (dark blue). This grouping failure is a limitation in the clustering process (KMeans) to extract object assignments and assumes clusters have equal angular variance which need not hold true in practice. Similarly, grouping failures in rows 5 and 6, where blue oval and purple square (row 5) or light blue heart and cobalt square (row 6) are incorrectly grouped together despite showing enough phase specialization in each case.

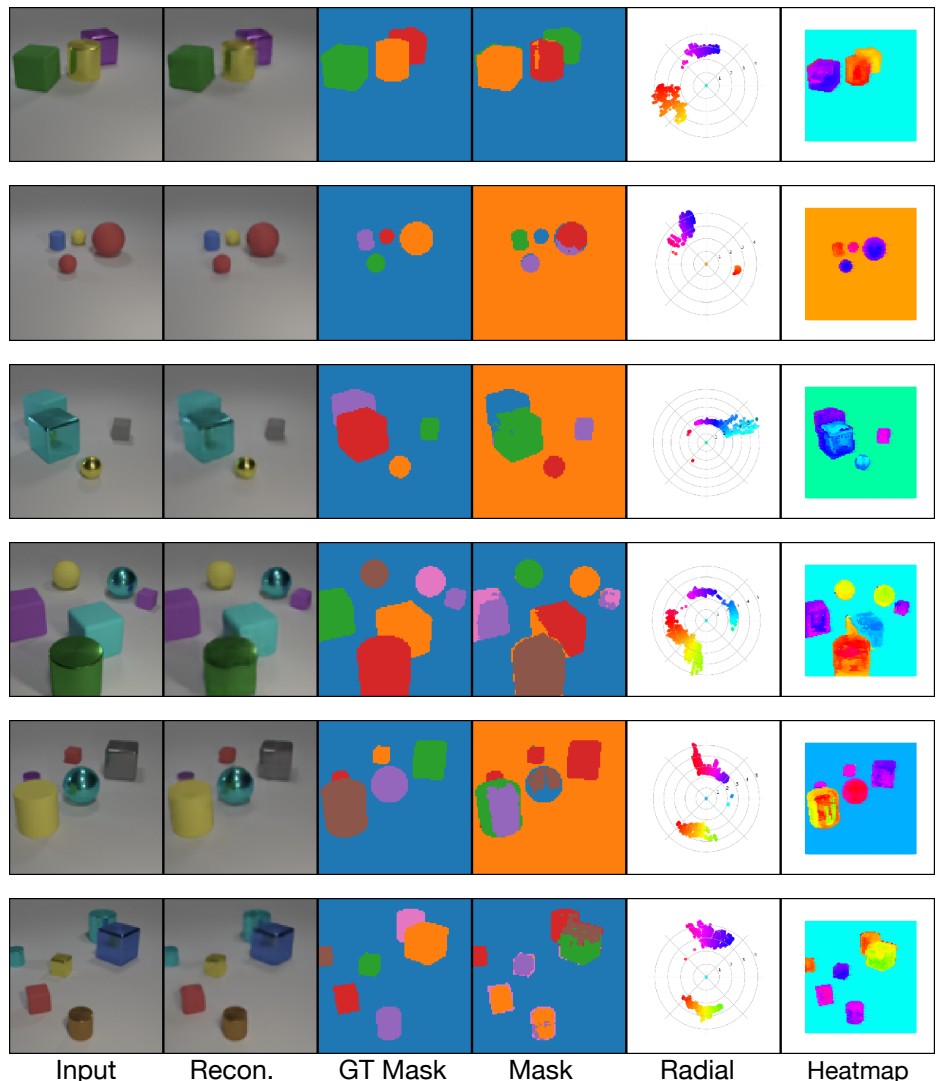

| Input | Recon. | GT Mask | Mask | Radial | Heatmap |

Figure 9: Collection of images grouped by SynCx from the CLEVR dataset. Rows 1-4 show cases where SynCx groups scenes containing three to six objects reasonably well. Row 5 shows a failure mode where the gray, red and purple objects are incorrectly grouped together despite showing noticeable (see column 6). This can be attributed to the limitations in the clustering process similar to the ones highlighted previously with regards to assumption of equal angular variance of clusters. Similar effects can be observed with the grouping errors in row 6.

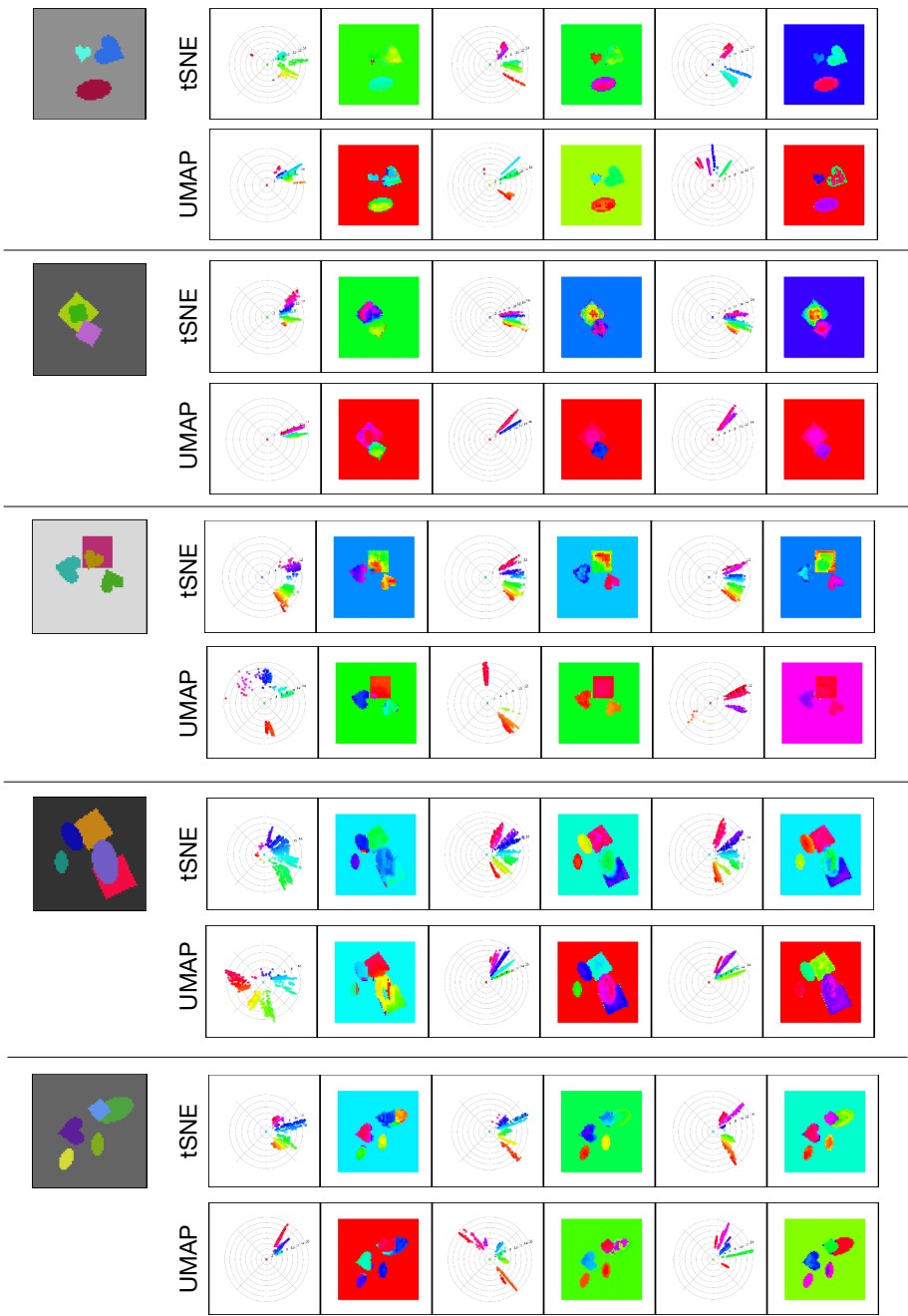

Figure 10: Collection of samples comparing the use of t-SNE versus UMAP for the dimensionality reduction computation in the visualization process. Panels 1 and 5 show cases where the clusters of projected phase maps of both t-SNE and UMAP correlate with the number of objects in the image. Panels 2, 3 and 4 show cases where the clusters of phases projected using t-SNE correlate with the number of objects in the image whereas the clusters of phases projected using UMAP do not. Overall, we find that t-SNE produces qualitatively better projections than UMAP, i.e., phase clusters correlate better to the number of objects in the image.

