# OpenReview forum: "Recurrent Complex-Weighted Autoencoders for Unsupervised Object Discovery"
_NeurIPS.cc/2024/Conference — NeurIPS 2024 poster_

### Official Review · Reviewer_3hk1 · 2024-07-09

**Soundness:** 2
**Presentation:** 3
**Contribution:** 2
**Rating:** 4
**Confidence:** 4

**Summary:**

The authors propose a new way to obtain binding through complex-valued activations in an autoencoder for object discovery. They introduce recurrence in the generated phase map and perform the operations using complex weights.
Their model reaches competitive performance on a SOTA benchmark for unsupervised object discovery, including datasets such as Tetrominoes, Multi dSprites, and Clevr, in terms of reconstruction and object separation.

**Strengths:**

The paper is very clearly written and nice to read.

The authors propose an interesting addition to existing complex-valued autoencoders in the recurrent dynamic of the phases. This approach is well motivated by neuroscience and shows a clear improvement in the object separation metrics.

The control experiment on greyscaled images highlights an interesting bias in existing models, which seems partially alleviated in the current model.

**Weaknesses:**

**The authors need to provide a more solid justification for the use of complex weights:** The previous models use real weights, as justified by Reichert & Serre, 2017 to align with neuroscience evidence. Indeed, the conjunction of multiplication by real weights followed by a sum of the complex activity is suggested as a natural inductive bias for synchrony: a complex addition between complex numbers with desynchronized phases will result in reduced activity in the amplitude. Conversely, in-phase activity will be enhanced, serving as an elegant way to reach synchrony in the population.
Applying complex weights will give more degrees of freedom to modify the phase value of each neuron, but it leads to less control in terms of resulting phase value: they can be potentially shifted to any angle, even opposite to their initial values. This behavior does not seem to align with neuroscience evidence, contrary to what the authors claim.

Related to these points, the authors claim, in lines 86 and 96,  that complex weights are *critical* to detect particular phase arrangements. But this behavior can potentially also be obtained with real-valued weights. In fact, the weights are applied to the real and imaginary parts of a complex activity. The phase arrangement is therefore present in the activity, and can also play a role in the detection of such features. This observation is even more valid when a specific phase bias is applied separately to the phase after the convolution, as proposed by Lowe et al, 2022.

**The evaluation pipeline of object assignment and phase map visualization differ and highlight a lack of good binding abilities:** The recurrence on the phases dynamic is supposed to enhance phase separation at the last iteration and in the last layer of the decoder. However, the authors mention that the object assignment (L177) and the phase map visualization (L212) come from *an appropriate/intermediate layer of the decoder*. Does this mean that the phases of the last layer of the decoder do not exhibit grouping properties? And for each model and dataset, do the object assignment and the phase map result from the same layer, or can they result from different layers even though they are from the same model and dataset? And why do the clustering metrics differ between object assignment (k-means) and phase map (t-SNE)? These results should be at the center of the binding abilities of the model, and all these differences seem to highlight a lack of clear ability of the model to perform such grouping. Ideally, all the metrics should be computed at the last iteration and in the last layer of the decoder. But if there is confound related to a color bias in the amplitude, the object assignment and the phase map should at least results from the same clustering algorithms and the same decoder layer.
This choice of layer is particularly critical since the grouping metrics are applied on the last layer of the decoder for all other baseline models.

**Are the baselines 100% comparable with SynCx?** The introduction of complex weights doubles the number of parameters compared to baseline models using real weights. Are the baseline models still comparable in terms of number or parameters? If not, how could the authors explain that the MSE loss of SynCx is always higher compared to RF and CtCAE (except on dSprites)?
Also, Slot-based models are mentioned in the related work but their performance on this benchmark is not included in the results. It would also be interesting to evaluate whether they might also be biased towards colors by reproducing the results of Table 5 with this family of models.

**Minor weaknesses**:
- Missing citation: even though modReLU was mentioned in Trabelsi et al, 2018; the modReLU was initially proposed by Arjovsky et al., 2015.
- The authors use the term "robustness" (L55) to express the excessive reliance of models on color cues. I'd suggest replacing this term as robustness represents something different in the DL literature.
- L100: parentheses missing after citation 24.

**Questions:**

- The example in Figure 1 is used to motivate the need for recurrence and complex-valued weights; have the authors tried to evaluate their model on similar stimuli to confirm that it can group the objects accordingly?

- The recurrent dynamic is done only on the phase: at each iteration t, only the phase at t-1 is fed as an input, and not the magnitude (the magnitude remains the input image at each iteration). Have the authors tried a closed-loop recurrence where the whole complex output at iteration t-1 constitutes the input at iteration t? This solution would seem more elegant and more in line with neuroscience as it is unlikely that the phase and firing rate of each neuron can be split in such a way.

- The initial modReLU equation seems to include a normalization of the amplitude by z/|z|. Do the authors include it as well, and if yes, can they add this term to equation (2)? If not, can the authors explain why not include this term? Also, is there any constraint applied to the bias to prevent the amplitude from being negative? If not, do the authors make sure the amplitude remains positive or switch the phase value accordingly?

- Table 4 shows the benefit of von-Mises initialization compared to Uniform. Have the authors tried to initialize the phases at zero, as done in the CAE?

- The colors in the polar plot visualization are supposed to reflect the phase value matched with the colors in the phase maps. However, why do some phases of similar angles have different colors? For example, in Fig 8, row 5, the yellow phases are represented close to angle 0, but in row 2 this same color represents phases close to -pi/2. Is it possible to use a standard colormap applied to each plot to visualize the phase distribution and compare between images?

- The t-SNE to visualize the phase map first ensures that the magnitude of the activity is set to 1 (L562). However, the result from the clustering exhibits an activity with magnitude differing in their amplitude value. Why is this the case?

**Limitations:**

The limitations focus only on the reliance of SynCx on color cues, even though this model performs better than the baseline in this experiment.

Some additional limitations should be included in the limitation section:
- Even though this model does not need an additional objective to perform binding, it employs twice as many parameters as the existing baselines. Additionally, the number of flops is also doubled and then multiplied by the number of iterations needed. This results in a model requiring significantly more computational resources than the baselines, to reach performance almost on par with RF and CtCAE.
- The weaknesses of the evaluation procedure should also be mentioned in the limitation section.

---

> ### Author Rebuttal · Authors · 2024-08-06
>
> We thank the reviewer for their feedback on our work and appreciate the positive feedback on the quality of our presentation.
>
> ``` But this behavior can potentially also be obtained with real-valued weights. In fact, the weights are applied to the real and imaginary parts of a complex activity. The phase arrangement is therefore present in the activity, and can also play a role in the detection of such features. ```
>
> Please note that simply using shared real-valued weights and applying it to compute matrix-vector products with real and imaginary parts of a complex-valued activation followed by summation alone does not lead to phase synchronization in the complex-valued activations in previous models (CAE++, CtCAE and RF). Phase synchronization in these models crucially relies on the use of an additional inductive bias i.e. the gating mechanism ($\chi$-binding) as noted by us (see lines 46-50) and echoed by Loewe et. al [23, 25]. Please refer to Table 2 in [23] and Appendix section D.3 in [25] where they perform an ablation on CAE and RF models without the $\chi$-binding mechanism. This ablated model variant fails to phase synchronize towards objects. Lastly, regarding our usage of the word “critical” we mean that the use of complex-valued weights is a natural choice to instantiate this function in our formulation of synchrony outlined (Introduction section). We do not mean that this is the only way to achieve phase synchronization behavior in general. We’re happy to adjust the corresponding text to avoid any misconceptions.
>
> ``` This behavior does not seem to align with neuroscience evidence, contrary to what the authors claim. ```
>
> We would like to clarify that our work does not claim that the SynCx model design and training procedure are biologically plausible. Instead we motivate the use of complex-weights and recurrence from a computational perspective as follows.
>
> Using real valued weights allow you to activate a hidden feature if two input features are aligned in phase, but you may wish to activate a hidden feature that indicates that the two input features should be grouped apart (e.g., 180 degrees out of phase).  This is a principled computational motivation for using complex weights. As for whether complex valued weights have biological plausibility, it's easy to think of mechanisms that effectively behave like complex-valued weights.  The critical mechanism is to detect features operating at different relative phases, which in the biological case, will correspond to different firing times.  Dendritic tree connectivity can implement synchrony that is achieved at arbitrary phase lags. We don't emphasize this in our paper (and certainly can, if the reviewer would find it helpful) because we are focused on showing the computational improvements and elegance that results from complex-valued weights. We hope this resolves any misinterpretations of our motivations.
>
> ``` And for each model and dataset, do the object assignment and the phase map result from the same layer, or can they result from different layers even though they are from the same model and dataset? ```
>
> Please note, that for grouping performance (ARI scores) of our model in all Tables (across all datasets) in the paper, we always cluster the phases from the penultimate decoder layer at the last iteration N. Similarly, for all phase map visualization we apply t-SNE to perform dimensionality reduction to the d-dimensional phases from the penultimate decoder layer. In Figure 4, we simply showed how these phases evolve across iterations. The visualization of phase maps (Figure 4 and associated text) is simply a qualitative tool for inspection. We do not use it to make any claims about the grouping performance of our model against baselines. We will edit the corresponding lines in text to make this point clear and avoid any confusion.
>
> ``` And why do the clustering metrics differ between object assignment (k-means) and phase map (t-SNE)?```
>
> Please note, that the visualizations of phase maps (Figure 4) are simply the 1D projections (dimensionality reduction done using t-SNE) of 64-D continuous phase vectors at each location in the spatial feature map. The visualization procedure does not apply any clustering procedure. The 1-d projected phase maps shown in Figure 4 are not similar to the discrete object segmentation masks (like columns 2, 4, 6 in Figure 3).
>
> ``` Slot-based models are mentioned in the related work but their performance on this benchmark is not included in the results. ```
>
> Thank you for this suggestion, this was noted by another reviewer as well. We will add the following object discovery results of SlotAttention to Table 1 in the updated version to provide more context. Slot-based models do not show this color bias on these datasets tested here.
>
> | Model         | Tetrominoes    | dSprites       | CLEVR          |
> |---------------|----------------|----------------|----------------|
> | SlotAttention | 0.995 +- 0.002 | 0.913 +- 0.003 | 0.988 +- 0.003 |
>
> We will also discuss this performance gap between SlotAttention and state-of-the-art synchrony-based models in the limitations section.
>
> ``` Missing citation: even though modReLU was mentioned in Trabelsi et al, 2018; the modReLU was initially proposed by Arjovsky et al., 2015. ```
>
> We thank the reviewer for this correction, we will update the citation as suggested.
>
> ``` The authors use the term "robustness" (L55) to express the excessive reliance of models on color cues. I'd suggest replacing this term as robustness represents something different in the DL literature. ```
>
> We thank the reviewer for pointing this out. We will use another term to avoid any misinterpretation. We would like to ask if the reviewer has any suggestions for alternatives?
>
> ``` L100: parentheses missing after citation 24. ```
> We thank the reviewer for this correction and will update it.
>
> end of part (1 / 3)

---

> ### Author Response · Authors · 2024-08-06
> **Rebuttal Response -- (part 2 / 3)**
>
> ``` The example in Figure 1 is used to motivate the need for recurrence and complex-valued weights; have the authors tried to evaluate their model on similar stimuli to confirm that it can group the objects accordingly? ```
>
> Please note that this illustrative example was intended to essentially serve as a pedagogical tool to explain the essence of the idea and provide intuition to the readers. All our experimental benchmarking is done on standard datasets of interest to the object-centric learning community. We have not trained/tested our model on such specially constructed types of stimuli.
>
> ``` The recurrent dynamic is done only on the phase: at each iteration t, only the phase at t-1 is fed as an input, and not the magnitude (the magnitude remains the input image at each iteration). Have the authors tried a closed-loop recurrence where the whole complex output at iteration t-1 constitutes the input at iteration t? This solution would seem more elegant and more in line with neuroscience as it is unlikely that the phase and firing rate of each neuron can be split in such a way. ```
>
> The process of grouping needs to be carried out on the input image. This grouping process iteratively refines a noisy initial phase map based on the features (magnitudes) and their associated bindings (phases).  Analogously, if we see the iterative refinement process of the SlotAttention module, the recurrence is applied only on the slots and not on both the (slots, inputs).
> Purely, out of curiosity during early prototyping we did try the version the reviewer has described and found it did not work well. Lastly, we would like to reiterate that our design choices for architecture and training procedure are not intended to be biologically plausible.
>
> ``` The initial modReLU equation seems to include a normalization of the amplitude by z/|z|. Do the authors include it as well, and if yes, can they add this term to equation (2)? If not, can the authors explain why not include this term? Also, is there any constraint applied to the bias to prevent the amplitude from being negative? If not, do the authors make sure the amplitude remains positive or switch the phase value accordingly? ```
>
> Please note the two ways of computing the modReLU activation are equivalent (see below). Therefore, this term has been included already by us.
>
> Using the same notation across our paper and Arjovsky et. al,
> Given,
> $y \in \mathbb{C}^d $ \
> $ | y | \in \mathbb{R}^d \rightarrow $ magnitude-component of complex-valued $y$ \
> $ \phi_y \in \mathbb{R}^d \rightarrow$ phase-component of complex-valued $y$ \
> Our version (in equation 2) is written as, \
> $ h = ReLU( |y| + b) * e^{i \phi_y} $ \
> Original modReLU from Arjovsky et. al (equation 8 in their paper) can be written as, \
> $ h = ReLU( |y| + b) * \dfrac{y}{|y|} $ \
> $ y = |y| * e^{i \phi_y} $ (Euler’s formula, as shown on the right side of equation 2 in our paper)
> Substituting this expression for y in the original modReLU we get, \
> $ h = ReLU( |y| + b) * \dfrac{|y| * e^{i \phi_y} }{|y|} $
> Canceling out the |y| terms in the numerator and denominator we get, \
> $h = ReLU( |y| + b) * e^{i \phi_y}$ (equal to our version of modReLU)
>
> In short, the modReLU activation function ensures that the magnitude of the output always remains strictly non-negative. Further, since the ReLU function is only applied on the magnitude component of $y$, applying the activation preserves the phases across the pre-activation ($y$) and post-activation ($h$) complex-valued vectors (no flipping). Please note that the magnitude-component of output complex-valued activation $h$ is obtained by applying the modReLU activation function only on the magnitude component of $y$ ($|y|$). The magnitude component of $y$ ($|y|$) is strictly non-negative and the activation function is also strictly non-negative. Therefore the magnitude-component of $h$ ($|h|$) will always remain strictly non-negative.
>
> ``` Table 4 shows the benefit of von-Mises initialization compared to Uniform. Have the authors tried to initialize the phases at zero, as done in the CAE? ```
>
> Please find the results using “zero” initialization for phases above. We can see that it performs comparably to the uniform case but worse than Von-Mises. This result is in line with our expectations. The best performing model variant (Von-Mises) samples “noisy” initial phases with an intermediate level of variance (Uniform -> very high variance, Zero -> no variance).
>
> | Phase Init. | MSE                | ARI          |
> |-------------|------------------|--------------|
> | Zero          | 2.46e-3 +- 5e-7 | 0.74 +- 0.04 |
>
> end of part 2/3

---

> ### Author Response · Authors · 2024-08-06
> **Rebuttal Response -- (part 3/3)**
>
> ``` The colors in the polar plot visualization are supposed to reflect the phase value matched with the colors in the phase maps. However, why do some phases of similar angles have different colors? Is it possible to use a standard colormap applied to each plot to visualize the phase distribution and compare between images? ```
>
> Please note, that the colors in the phase maps are matched (between radial plot and heatmap) only within a single row and not across rows. Regarding the standardization of colormap across different image samples (rows), we have not managed to achieve that. We understand that having uniform colors across all rows would be nicer. We would be happy if the reviewer can give us some pointers for the same? Thanks for your feedback on this!
>
> ``` The t-SNE to visualize the phase map first ensures that the magnitude of the activity is set to 1 (L562). However, the result from the clustering exhibits an activity with magnitude differing in their amplitude value. Why is this the case? ```
>
> Thank you for bringing up this point about the magnitudes in the radial plot visualizations. Please note that our phase map visualization is an extension of similar plots in prior works such as CAE[23] (see Figure 6 in [23]). In their plots we can see in the radial plots the true magnitudes are shown along with colors that match the orientations (phases) from the heatmap. We have tried to maintain stylistic consistency in the radial plots in [23] and ours.
>
> ``` limitations focus only on the reliance of SynCx on color cues, even though this model performs better than the baseline in this experiment. ```
>
> This experiment highlights a flaw in the general class of synchrony-based models. Please note that our model performance also drops on the grayscale CLEVR compared to the original colored version (shown in Table 1). We believe this is a valuable insight that poses an open problem in this class of synchrony-based models.
>
> ``` It employs twice as many parameters as the existing baselines. Additionally, the number of flops is also doubled and then multiplied by the number of iterations needed. This results in a model requiring significantly more computational resources than the baselines, to reach performance almost on par with RF and CtCAE. ```
>
> Please note that In fact, our model has far fewer parameters compared to all competing baselines. For the models shown in Table 1, our model is much smaller than competing baselines by a factor between 6-23x. This further highlights the computational benefits of our simple ideas of complex-valued weights and recurrent computation.
>
> Please find the model parameters for all models shown in Table 1 is listed below. Model parameters (rounded up to the nearest thousand) for all results shown in Table 1. Total number of model parameters for all models expressed in terms of number of real-valued floats.
>
> | Model | Tetrominoes | dSprites   | CLEVR      |
> |-------|-------------|------------|------------|
> | CAE++ | 5,372,000   | 11,713,000 | 22,281,000 |
> | CtCAE | 5,372,000   | 11,713,000 | 22,281,000 |
> | RF    | 6,630,000   | 11,861,000 | 22,592,000 |
> | SynCx | 966,000     | 974,000    | 974,000    |
>
> Please find the wall-clock times (rounded up to the nearest minute) for training for our model and the best performing baseline model RF on Tetrominoes and dSprites to achieve the same results as shown in Table 1. Both models have been trained using the exact same hardware setup (1 NVIDIA P100 GPU).
>
> | Model | Tetrominoes | dSprites |
> |-------|-------------|----------|
> | RF    | 7h 20m      | 4h 50m   |
> | SynCx | 1h 35m      | 4h 10m   |
>
> From the above table we can see that SynCx converges much faster than RF on Tetrominoes where it also significantly outperforms RF in terms of ARI scores. On dSprites SynCx’s grouping performance is comparable to RF and both models take comparable times to converge. We thank the reviewer for bringing up these concerns. We hope that the additional information here resolves their concerns regarding the fairness of comparison of our model against baselines. We will highlight these points about our model and baselines in the updated version.
>
> ``` Are the baselines 100% comparable with SynCx? Are the baseline models still comparable in terms of number or parameters? ```
>
> We believe that the comparison of SynCx with baselines is fair. Notably, our model has far fewer model parameters than competing baselines as shown above.
>
> ``` If not, how could the authors explain that the MSE loss of SynCx is always higher compared to RF and CtCAE (except on dSprites)? ```
>
> Since our model has far smaller (weights) compared to the baselines and so it is reasonable to expect higher reconstruction loss for our model compared to RF and CtCAE. However, we would like to emphasize that the end goal of training this model is not to achieve better image reconstruction but learning better representations, i.e. phase synchronization towards objects.
>
> end of part 3/3

---

> > ### Comment · Reviewer_3hk1 · 2024-08-13
> >
> > I thank the authors for answering my comments.
> >
> > It is however still unclear to me why different procedures are used for visualization and clustering. The authors mentioned t-SNE clustering for visualizations (L216) but mentioned in their rebuttal that "The visualization procedure does not apply any clustering procedure".
> > As for aligning the colors for all the images, I would simply suggest to the authors to remap the phases to the same range (i.e. -pi and pi). The projection on the unitary circle will consequently be comparable on all the plots.
> >
> > It is interesting that SynCx is more efficient in terms of parameters and training time. Could the authors add details of the architecture of the Auto-Encoder to the Appendix?
> >
> > Overall I agree with Reviewer Ta5A: the paper should contain more scientific insights about perceptual grouping. As it is presented now, it feels like a list of technical details leading to these specific results.
> > For this reason, I doubt the generalizability of the proposed solution and I keep the current score unchanged.

---

> > > ### Author Response · Authors · 2024-08-13
> > > **rebuttal discussion**
> > >
> > > Thank you for the response. Please note that we use t-SNE to perform non-linear dimensionality reduction and not clustering. L216 says: “We apply t-SNE [38] to perform dimensionality reduction on the complex-valued feature map ….”). Our phase map visualization procedure generalizes analogous visualizations in prior work (see Figure 4 in CAE [25]) for the case of  multi-channel phases at every location in the feature map.
> > >
> > > Regarding the architecture details of our model, please find the encoder and decoder modules of our model in Appendix A. We will add the Tables comparing parameter counts and training times of our model against baselines in the Appendix as well. Our detailed responses and additional experiments in the rebuttal have been addressed to resolve your primary concerns expressed in the review regarding fairness of comparisons against baselines (parameter counts, training times) and clarify the motivation for use of complex-weights.
> > >
> > > Thank you for your suggestion to focus more on the aspects of perceptual grouping and less on the technical details. In our paper (and rebuttal response) we have tried to do that. We performed these ablations to dissect the impact of individual components and make sure there are no non-essential ones (i.e. they all contribute positively to the phase synchronization). Lastly, we would like to know if the reviewer has actionable recommendations towards "more scientific insights about perceptual grouping" we could pursue?

---

> > > > ### Comment · Reviewer_3hk1 · 2024-08-14
> > > >
> > > > The k-means clustering was not used in the CAE [25] for phase visualization. Figures 3 and 4 show the raw phase value without any additional clustering or dimensionality reduction method. Clustering was used in [25] to provide some quantitative measures of phase separation and evaluate the quality of the representation learned by the model.
> > > > Performing clustering prior to the visualization in this current paper suggests that the phases fail at synchronizing to represent the different objects and that clustering techniques have to be employed to induce some notion of phase separation.
> > > > This point makes me doubt the generalizability of the model to other more realistic scenarios.
> > > >
> > > > Additionally, I find it interesting that the current architecture proposed by the authors contains fewer parameters than the baselines. However, this model does not always outperform all the baselines in terms of reconstruction and object separation.
> > > > I, therefore, wonder why the authors have not tried to add additional layers to the model to better compete with these baselines and would consequently like to see the performance that a model with a matching number of parameters could reach and whether it would significantly outperform the current baselines.

---

### Official Review · Reviewer_sSNp · 2024-07-13

**Soundness:** 3
**Presentation:** 3
**Contribution:** 2
**Rating:** 6
**Confidence:** 3

**Summary:**

The paper aims to solve the problem of unsupervised object segmentation, which can be viewed as a feature binding problem when feature detectors process images at different locations in parallel. It proposes an improvement to synchrony-based approach which uses complex numbers as activation value for neurons: the weights are also complex numbers. In addition, the model processes images in recurrent way and let phases for neurons at different locations of an object incrementally group together. It also finds that a representational bottleneck in an auto-encoder architecture is required for the model to work.

**Strengths:**

The paper illustrated superior or competitive performance on several common datasets against other synchrony-based binding algorithms for unsupervised object segmentation.

The design of complex weights are well motivated and makes perfect sense for a reader after reading it.

It gets rid of contrastive training, $\chi$-binding, and other tricks used by similar models.

**Weaknesses:**

I appreciate that the authors acknowledges that the model still relies on color cues to group pixels to a large degree, which is true to previous models such as RF, and its limitation in dealing with metallic or matte texture. It is possible that this is a fundamental problem for unsupervised object detection model that overly relies on a bottleneck to group pixels, as shown by https://arxiv.org/abs/2403.03730. Color is simply the easiest cue when a bottleneck is enforced, but there is no drive for such a model to learn cues relevant to real-world scenarios with complex lighting conditions and textures. Therefore, it is unlikely for solution proposed in the current model to be an ultimate solution for object perception. One may even ask whether the performance is mainly gained by the bottleneck or by the phase synchrony being proposed. To answer this, it seems that a comparison with earlier models not in the synchrony-based binding family, such as MONet or slot-attention might provide a clue. But such comparison is unfortunately not provided.

Nonetheless, I think the complex weight idea is a good addition to the subdomain of synchrony-based binding models.

Another limitation is that the detection of object relies on post-hoc clustering of phase. So it remains to be seen how it works when there are many objects, i.e., whether the $2\pi$ space of phase is sufficient to separate different objects.

Lastly, if we think about the brain, especially of animals without language, their ability of learning object detection in unsupervised way should ultimately help them develop some sort of invariance: the same object should be represented similarly across different situations such that a squirrel will know that a new acorn is similar to other acorns and should still edible, unlike a plastic bottle. Although the invariance has been less focused by most of the object-centric models, it can be interesting to evaluate whether the phase is similar for similar object across images.

**Questions:**

In the demo for dSprite in Fig 4, it appears that the phases are already well separated after the 1st iteration, but got more spread out in the second iteration and got back better in the third iteration. So I wonder whether such fluctuation in clustering is common.

**Limitations:**

The authors properly addressed an important limitation of the model, although I think there are some more as I wrote in the weakness. There is no negative societal impact as I can see.

---

> ### Author Rebuttal · Authors · 2024-08-06
>
> We thank the reviewer for their support of the motivation behind the use of complex weights and its rationale. We thank them for their support of the simplicity of our model which eliminates the need for contrastive training, $\chi$-binding and other heuristics.
>
> ``` I appreciate that the authors acknowledges that the model still relies on color cues to group pixels to a large degree, which is true to previous models such as RF, and its limitation in dealing with metallic or matte texture. ```
>
> We thank the reviewer for the positive comments about the honesty in our evaluation of current state-of-the-art synchrony models. We would like to emphasize that our method (SynCx) relies far less on solely color cues compared to current state-of-the-art synchrony models such as RF (see lines 285-287).
>
> ``` Another limitation is that the detection of object relies on post-hoc clustering of phase. So it remains to be seen how it works when there are many objects, i.e., whether the  space of phase is sufficient to separate different objects. ```
>
> In our experiments we show that our model is capable of handling up to 5 or 6 objects (see Figure 8 row 3 and Figure 9 row 4). Note, that the phase space is still not fully covered indicating that it could potentially accommodate more objects. We will comment on this as well in the limitations section.
>
> Regarding the post-hoc phase clustering, this ability of synchrony-based models to maintain a continuous phase representation for binding adaptable by gradient-based learning can be viewed as a strength compared to slot-based models where the binding information, i.e. number of slots and slot capacity are set apriori and not adaptable by gradient-based learning. This makes it difficult for slot-based models to adaptively change the coarseness of grouping at test-time without more fine-tuning or even re-training from scratch. Whereas our model (and synchrony-based models in general) maintain a continuous representation of object bindings adaptable by gradient-based learning. And we can post-hoc get different groupings with varying levels of granularity by modulating the clustering procedure applied on the continuous phase map. This gives the phase representation produced by synchrony-based models greater flexibility in chunking up the input image.
>
> ``` it is unlikely for solution proposed in the current model to be an ultimate solution for object perception. One may even ask whether the performance is mainly gained by the bottleneck or by the phase synchrony being proposed. ```
>
> We appreciate the comments from the reviewer. We agree with the reviewer that we don’t believe that the contributions presented here are the ultimate solution and we have not made such claims in our work either. Instead, we advocate for a simpler approach (lesser inductive biases and simpler training procedure) compared to all other synchrony-based models (CAE, CAE++, CtCAE and RF).
>
> ``` it seems that a comparison with earlier models not in the synchrony-based binding family, such as MONet or slot-attention might provide a clue. But such comparison is unfortunately not provided. ```
>
> Thank you for this suggestion. We acknowledge the gap in unsupervised grouping performance between current state-of-the-art synchrony-based models and slot-based approaches such as SlotAttention as noted by the reviewer. We will add the following object discovery results of SlotAttention to Table 1 in the updated version to provide more context.
>
> | Model         | Tetrominoes    | dSprites       | CLEVR          |
> |---------------|----------------|----------------|----------------|
> | SlotAttention | 0.995 +- 0.002 | 0.913 +- 0.003 | 0.988 +- 0.003 |
>
> It is noticeable how the performance gap between recent synchrony-based models (esp. SynCx) and SlotAttention has been narrowing on Tetrominoes and dSprites. We will also discuss this performance gap between SlotAttention and synchrony-based models on CLEVR dataset in the limitations section. Synchrony-based models have received far less attention by the research community and are making strides very recently to bridge the gap to predominant slot-based models which have been widely improved by the community efforts over a longer timeframe. Therefore, we view synchrony-based models as an exciting research direction with open problems to solve.
>
> ``` Although the invariance has been less focused by most of the object-centric models, it can be interesting to evaluate whether the phase is similar for similar object across images. ```
>
> This is an interesting question, regarding representational invariance for an object across viewpoints. When using synchrony-based models, what matters for separating objects is the relative phase (phase difference) between object phases rather than the absolute phase values assigned to pixels of any particular object. In fact, it’s plausible that different objects could take the same phase value (eg. $\dfrac{\pi}{3}$) across different iterations or when starting with different random initializations for the phase map. In the reviewer’s example different views of the acorn are not encoded by some absolute phase value (eg. $\dfrac{\pi}{4}$) thereby leading to invariant representations of the acorn.
> Lastly, we believe that this approach of using multiple views of an object for prediction provides useful training signals for an object-centric model. However, this is outside the focus (scope) of the research questions addressed here and an interesting avenue for future work.
>
> end of part 1/2

---

> ### Author Response · Authors · 2024-08-06
> **Rebuttal Response -- (part 2/2)**
>
> ``` In the demo for dSprite in Fig 4, it appears that the phases are already well separated after the 1st iteration, but got more spread out in the second iteration and got back better in the third iteration. So I wonder whether such fluctuation in clustering is common. ```
>
> We appreciate the reviewer’s feedback on our phase map visualization. Please note, these visualizations of phase maps (in Figure 4) are not the same as the discrete segmentation masks (clustered phases) shown in columns 2, 4, 6 of Figure 3. In simple terms, the visualization technique in Figure 4 is used to project a high-dimensional phase vector (at every spatial location) to 1D by applying dimensionality reduction (t-SNE). Our visualization technique for phase maps extends similar techniques provided in prior models (see Figure 4 in CAE [23]) to high-dimensional phase features from hidden layers. We cluster the phase maps from the last iteration to extract discrete segmentation masks for objects after the iterative process has settled to a relatively stable configuration. It is reasonable to expect that phase values will fluctuate (to some degree) before this process has settled as the phases are initialized with random (noisy) values.
>
> end of part 2/2

---

> > ### Comment · Reviewer_sSNp · 2024-08-12
> >
> > I thank the authors for addressing my comments. Although I am not someone chasing for benchmark, I am still not convinced that the synchronization-based approach adds much to solving the existing limitation of slot-based approach, especially given the performance gap. That being said, I appreciate the effort of continuing developing methods in a direction drawing less attention. Only for this sake, I increase my rating by 1.

---

> > > ### Author Response · Authors · 2024-08-13
> > > **rebuttal discussion**
> > >
> > > Thank you for the response and encouragement. We believe that classes of synchrony-based and slot-based models need not only be viewed as direct competitors. They could serve as complementary solution strategies for overcoming the binding problem with artificial neural networks.

---

### Official Review · Reviewer_nb8Q · 2024-07-14

**Soundness:** 3
**Presentation:** 4
**Contribution:** 3
**Rating:** 7
**Confidence:** 4

**Summary:**

The paper introduces SynCx, a fully convolutional auto-encoder with complex values that is applied iteratively. This recurrent model takes advantage of the ability of complex-valued weights to not only compute features but also bind similar features that can be encoded in the phase, creating a grouping effect. This approach enhances previous attempts at using synchrony to solve the binding problem by employing recurrence to propagate local knowledge. This explains its success in circumventing previous issues with synchrony that relied on shortcuts such as color to solve the problem.

**Strengths:**

The paper offers a clear and precise introduction that allows the user to understand the method but also the rationale of the choices taken. The method albeit simple, seems to provide an effective way to advance the state of the art of the binding problem. There are qualitative and quantitative evaluations that well support the main claims of the paper.

**Weaknesses:**

It has been shown that TSNE have some limitations, specifically results can change dramatically based on the choice of iterations and perplexity. How were these values selected? Also perhaps good using another dimensionality reduction technique to perform the visualization such as  UMAP.

**Questions:**

Maybe a minor question, in figure 4 there is a visualization of the iterative process, showing how every iteration seem to provide a better grouping result. However, in the phase step is not clear to me how every step is contributing to the propagation of the local context.

**Limitations:**

Maybe good to add a sentence clarifying the limitations when visualizing the process.

---

> ### Author Rebuttal · Authors · 2024-08-06
>
> We thank the reviewer for their positive comments regarding the simplicity of our model, the ease of understanding the rationale behind our design choices and for noting that the evaluations support the main claims in our paper.
>
> ``` It has been shown that TSNE have some limitations, specifically results can change dramatically based on the choice of iterations and perplexity. How were these values selected? Also perhaps good using another dimensionality reduction technique to perform the visualization such as UMAP. ```
>
> We thank the reviewer for these helpful recommendations regarding our phase visualization process. Regarding the t-SNE hyperparameters used (listed in lines 564-566) we tried out a few values for n_iters=[250, 500, 1000] and perplexity=[15, 20, 30] in the prototyping phase. We settled on the values of n_iter=500 and perplexity=20 for the visualizations shown in the paper since they gave good initial results (visual inspection) within a reasonable amount of processing time per image. During the prototyping phase, we compared UMAP against t-SNE for performing dimensionality reduction in our visualization process (download PDF from https://file.io/gMEMsZWRKNZX). We used the default UMAP hyperparameters and t-SNE hyperparameters listed above for this comparison. We find that t-SNE produces qualitatively better projections than UMAP, i.e., phase clusters correlate better to the number of objects in the image (see panels 2,3 and 4). Lastly, the visualization process takes around 8 seconds per image with t-SNE compared to 24 seconds per image for UMAP. Therefore, we decided that t-SNE is the better choice for our use case here.
>
> ``` Maybe a minor question, in figure 4 there is a visualization of the iterative process, showing how every iteration seem to provide a better grouping result. However, in the phase step is not clear to me how every step is contributing to the propagation of the local context. ```
>
> Thank you for your question. We agree with the reviewer that it is difficult to visually interpret from sequences of phase maps and radial plots to see how local context gets propagated. The illustrative example in Figure 1 was meant to serve as an idealized mode of operation of such a system. Regarding a way to visually show how local context is propagated. We think a possible approach would be to visualize/inspect local activation patches to show some measure of ‘agreement’ in their phases across successive iterations. Currently, we do not know a simple visualization method to visually show this phenomenon. We are curious to know if the reviewer has some recommendations for the same.

---

> ### Comment · Reviewer_nb8Q · 2024-08-11
>
> I want to thank the authors for the responses and the t-SNE experiments. It does seem that the UMAP has a harder time showing the separation of some  objects. For instance, in the third one it may even lead to a failure mode. Perhaps a good point to add to the limitations. For the second question I wonder if a the difference between consecutive steps would provide the regions with higher change, and perhaps there is a connection there, but I agree with the authors that this may be hard to visualize in a simple manner. I will increase my score one point, but looking forward to discuss with the other reviewers and see how the other threads evolve.

---

> > ### Author Response · Authors · 2024-08-13
> > **rebuttal discussion**
> >
> > Thank you for the feedback and revision of the score. We will comment on the UMAP results.

---

### Official Review · Reviewer_g2JM · 2024-07-15

**Soundness:** 3
**Presentation:** 3
**Contribution:** 3
**Rating:** 7
**Confidence:** 4

**Summary:**

Summary:

The authors introduce SynCx, a fully convolutional recurrent complex-valued autoencoder (AE) designed for unsupervised object discovery. SynCx operates by iteratively refining a randomly initialized phase array, which is the same shape as the input image, to reconstruct the input image. At each stage, the AE decoder produces a complex-valued output, where the magnitude represents the reconstructed image and the phase encodes object binding information. The decoder output is computed as a function of the input image, gated by the most recent phase (binding state). The model is trained using pixel-wise reconstruction error. Experimental results demonstrate that SynCx excels at unsupervised object discovery in comparison to relevant compared baselines and is especially good at binding in the presence of conflicting stimuli with identical low-level features.

**Strengths:**

Strengths:
- The proposed SynCx model comprises of a notably simple and elegant architecture + training. A key advantage over other unsupervised object discovery techniques is that SynCx does not rely on a fixed number of discrete object slots or buckets, allowing it to scale more flexibly to scenes with varying numbers of elements.
- The experimental results, presented with standard deviation, demonstrate the effectiveness of SynCx in unsupervised object discovery, outperforming relevant baselines (Table 1).
- The authors provide valuable insights through ablation studies on the bottleneck, number of iterations, and phase initialization distribution, validating the chosen configuration of SynCx.
- The writing is clear and concise, and the figures are well-organized and easy to understand, making the submission a pleasure to read and greatly contributing to its overall clarity.

**Weaknesses:**

Weaknesses:
- The paper's illustrations primarily demonstrate SynCx's performance on scenes with non-overlapping objects or overlapping objects with distinct low-level features (e.g., Figure 4 in dSprites). However, it is unclear how well SynCx would perform on scenes with overlapping objects that share similar low-level features (e.g., same color or shape). Specifically, would the phase representation effectively encode the presence of multiple distinct objects at overlapping pixels, mimicking human vision's ability to separate them?
- The paper does not provide clear guidance on determining the optimal number of recurrent iterations for running SynCx, particularly in the absence of ground truth object identity information. For instance, if the number of objects in the scene increases significantly during test time, would SynCx still perform effectively with the same number of iterations used during training, where the number of objects was lesser?
- While SynCx shows promise, it is uncertain whether the approach would generalize to binding on naturalistic stimuli. This limitation is shared with existing baselines too, but addressing this gap in future work would be a valuable extension of SynCx.

**Questions:**

- What happens if SynCx is run with more recurrent iterations than what was used during training? Is the model's binding encoding stable at RNN $t > t_{train}$, or does the object discovery performance degrade during this "extrapolation" phase?

**Limitations:**

The authors have adequately addressed limitations in this submission.

---

> ### Author Rebuttal · Authors · 2024-08-06
>
> We thank the reviewer for their positive comments and support for the simplicity and elegance of our model/training, quality of our experimental validation/analysis and the overall presentation. We’re happy that the reviewer enjoyed reading our work.
>
> ``` However, it is unclear how well SynCx would perform on scenes with overlapping objects that share similar low-level features (e.g., same color or shape). Specifically, would the phase representation effectively encode the presence of multiple distinct objects at overlapping pixels, mimicking human vision's ability to separate them? ```
>
> This is an interesting observation about the phase representation. In Figure 9 (row 3) you can see two blue cubes (same shape / similar color) which partially occlude each other and our model has achieved good grouping of the two objects. Looking at the clustered phase maps (‘Mask’ column 4) the discretized phase at the overlapping regions encodes the presence of the object in the front and does not encode the occluded object as well. Looking at the t-SNE reduced phase map (‘Heatmap’ column 6) also we see that overlapping regions encodes the presence of the object in front (shades of blue) and does not encode the occluded object in the back (shades of purple).
>
> Another example, in Figure 9 (row 1) you can see a yellow cylinder and purple cube both metallic (same texture) which partially occlude each other. Again we see the same behavior in the phase representation where in the overlapping regions has specialized towards the object in front rather than the occluded object at the back.
>
> ``` if the number of objects in the scene increases significantly during test time, would SynCx still perform effectively with the same number of iterations used during training, where the number of objects was lesser? ```
>
> That’s an interesting question. We suspect that what matters more for convergence is the density of objects. Because of the convolutional architecture, patches are processed in parallel, and more objects won’t necessarily be harder to resolve if they are dispersed in the scene. We would like to investigate convergence time as a function of number and density of objects, but unfortunately the rebuttal period was too short to provide you with a concrete answer.
>
> ``` While SynCx shows promise, it is uncertain whether the approach would generalize to binding on naturalistic stimuli. This limitation is shared with existing baselines too, but addressing this gap in future work would be a valuable extension of SynCx. ```
>
> We share the reviewers view on the scalability of this class of models to more visually complex scenes. We have shown the current limits of these synchrony-based models even on simple datasets like CLEVR which is far from perfect decomposition (ARI score of 1.0). We hope to continue research on these synchrony-based models to bridge this gap in future work. We view the contributions in this work as an initial template towards this end goal due to the simplicity of the model and its training procedure.
>
> ``` What happens if SynCx is run with more recurrent iterations than what was used during training? Is the model's binding encoding stable at RNN 𝑡>𝑡𝑡𝑟𝑎𝑖𝑛, or does the object discovery performance degrade during this "extrapolation" phase? ```
>
> Table below shows the object discovery performance for our model on dSprites when trained with 3 iterations and evaluated using 4, 5 and 6 iterations. We can see that there is no degradation in object discovery performance of our model when we extrapolate the number of iterations used at test time.
>
> | No. of Iterations | ARI          |
> |-------------------|--------------|
> | 4                 | 0.81 +- 0.01 |
> | 5                 | 0.82 +- 0.01 |
> | 6                 | 0.82 +- 0.01 |

---

### Official Review · Reviewer_Ta5A · 2024-07-17

**Soundness:** 2
**Presentation:** 3
**Contribution:** 2
**Rating:** 5
**Confidence:** 4

**Summary:**

The authors propose SynCx: a model of synchrony-based perceptual grouping. SynCx is a recurrent autoencoder with complex-valued parameters. By design, phase-information in the model is carried through across model iterations allowing it to be "stateful". SynCx demonstrates competitive/comparable performance on dSprites and CLEVR while outperforming currently available synchrony-based methods on Tetrominoes. Its worth noting that SynCx operates in a fully unsupervised manner and does not require sophisticated hueristics, or augmentations to its training objectives, or additional forms of supervision to perform object discovery.

**Strengths:**

This paper targets an interesting problem in perceptual science -- what are the cues (at varying levels of abstraction) that are required to execute successful feature binding to represent "objectness". They expose a problem in current state-of-the-art methods in their overreliance on a low-level cue: color. The authors perform empirical analysis on a color-lesioned version of CLEVR and find that SynCx's reliance on color is lesser compared to the other baseline. SynCx is comparable in performance to more sophisticated methods on two challenging benchmarks.

**Weaknesses:**

Despite the promise of this approach, there are several outstanding questions to be addressed by this manuscript.

L178 "We use latent-level complex-valued features ... than simply color cues (RGB space)": This seems contradictory to the authors primary claim. If only the phase information is used for the clustering analysis (and this is stateful) then it is unclear why the authors suggest the use of latents vs outputs? Can the authors clarify?

The color-lesion experiment on CLEVR has potential low-level confounds apart from color (like shadows, shading, etc.) that models can use as shortcuts for binding.

Why did the authors have to retrain SynCx on the grayscale dataset? If binding-by-synchrony is faithfully implemented by their model, shouldn't this work zero shot? If this is not the case, then what is the rationale for it?

The authors raise a point about instance-level groups being the focus of object-centric learning (L59). Given that, there are very few (maybe 2) examples presented where the scene contains multiple instances of the same object (same color, size, orientation, etc.) and the corresponding outputs from SynCx. Can the authors present these?

The "lesioning expermients" part of the manuscript needs more thorough work.

"Effect of Bottlenecks": It is understandable that the test loss is lower for the model without bottlenecks since it has a greater number of parameters. However, the explanation for why bottlenecks are important seems misleading. The importance of bottlenecks, in my opinion, originates due to the effective receptive field sizes in these layers and has nothing to do with complex-valued units or synchrony per se. It is unclear what this analysis adds to the manuscript.

"Effect of Iterations": From the results presented, it seems like there is diminishing returns after just 2 iterations? If that's the case, then wouldn't this approach be very comparable to feedforward methods? Moreover, can the authors train SynCx on a large number of timesteps (say 10) and observe the dynamics of convergence in the phases during test time? Claims about "iterative constraint propagation" needs further backing. Perhaps these datasets are too easy?

"Effect of Phase Initialization": I found it hard to grasp the take away of this section. The Von-Mises distribution (with mean 0, and concentration 1) biases most of the units in the initialization to have similar phases. The random initialization performed pretty poorly. What does this suggest though computationally?

Figure 2 is slightly misleading in its depiction. The magnitude of the output $\mu_z^1$ is shown to be the same as $\mu_x$. This derails the reader from understanding the flow of information across timesteps. Notationally, denoting time as a superscript and outputs as $z$s is confusing from a readers perspective. A clearer version could be something along the lines of $\hat{x}(t) = \mu_{\hat{x}}(t) e^{i\phi_\hat{x}(t)}$

Minor (typos):

Figure 5 caption: "Sync without the bottleneck"

L135: Grammatical. "... or strongly competitive"

**Questions:**

Please refer to weaknesses above.

---

> ### Author Rebuttal · Authors · 2024-08-06
>
> We thank the reviewer for acknowledging the simplicity of our approach which does not require sophisticated heuristics or additional training objectives or forms of supervision. We address your specific concerns below.
>
> ``` L178 "We use latent-level complex-valued features ... than simply color cues (RGB space)": This seems contradictory to the authors primary claim. If only the phase information is used for the clustering analysis (and this is stateful) then it is unclear why the authors suggest the use of latents vs outputs? Can the authors clarify? ```
>
> The short answer to your good question is that we tried to read out from the output phases but they were inferior to using latent phases to segment objects. We found that the richer latent-level phases associated with higher-order features are useful for grouping. We also experimented with variants of the model that used a single (shared) phase for all outputs, and where each output feature used its own phase. In the classic supervised learning approach to segmentation with complex-valued units (model trained to predict image and segmentation mask), we could cluster the output phases, but the training signal supported meaningful information in the output phases. We are continuing to try to understand why phase information is lost at the output in the fully unsupervised setting, but one reason is likely that a unidimensional phase representation is too restrictive. In support of this conjecture is the fact that, in RF, Lowe et. al [25] have incorporated multiple ‘phase’ (rotation) components for each neuron. Our alternative to this, is to use the high-dimensional phases from latent features. We will comment on these aspects in the updated version.
>
> ``` The color-lesion experiment on CLEVR has potential low-level confounds apart from color (like shadows, shading, etc.) that models can use as shortcuts for binding. ```
>
> While it is true that there are these low-level differences in the image, this is far less of a cue than color signal. Colors can be accessed directly from the pixel values whereas these other cues require integration of multiple pixels. In short, the color cues are a shortcut feature that has high availability (Hermann et al. 2024) because individual pixels can specify the output grouping. Removing these cues (shortcut features) allows us to evaluate how much the other models rely on them (and to show that ours does to a lesser extent). Additionally, shadows make it perhaps even harder to do correct binding, and shading inside one object (e.g. for shiny metallic objects) arguably also makes the task harder.
>
> On the Foundations of Shortcut Learning, Hermann et. al 2024.
>
> ``` Why did the authors have to retrain SynCx on the grayscale dataset? If binding-by-synchrony is faithfully implemented by their model, shouldn't this work zero shot? If this is not the case, then what is the rationale for it? ```
>
> The experimental setup as suggested by the reviewer would introduce a sort of test-time out-of-distribution evaluation that challenges all object-centric models, as illustrated by the fact that SlotAttention [33] which performed similar grayscale experiments required retraining from scratch (see Figure 4 in [33]). Since the statistics of features belonging to any object have changed due to the grayscale transformation it is reasonable to expect the previously learned binding mechanisms on the colored images requires retraining the model weights. As such, it would be quite unreasonable to expect current state-of-the-art synchrony or slot-based approaches (eg. SlotAttention [33]) to zero-shot be able to group grayscale versions if it was only trained on RGB counterparts. Lastly, this experiment on grayscale CLEVR is designed to test whether the models have the capability of leveraging other image statistics other than color. And we show that our model (SynCx) significantly outperforms previous synchrony-based models.
>
> ``` The authors raise a point about instance-level groups being the focus of object-centric learning (L59). Given that, there are very few (maybe 2) examples presented where the scene contains multiple instances of the same object (same color, size, orientation, etc.) and the corresponding outputs from SynCx. Can the authors present these? ```
>
> The point about instance-level groups was intended to highlight a commonly observed failure mode of state-of-the-art baseline model RF [25] which groups all instances of a semantic category such as ‘cow’, ‘train’, ‘bread slice’ or ‘banana’ as one group. This can be seen in Figures 2 & 20 in RF [25] where it groups different cows or bananas or trains as one entity. Note here, that the two cows or bananas or two bread slices are not exact copies (replicas) rather these object instances only partially share some features such as color, texture, shape as they belong to same semantic category. We can also see this failure mode of RF surfaces again on our datasets when it fails to disambiguate between two blue tetris blocks or two yellow cylinders in Figure 3. It primarily seems to use color as a shortcut feature to discriminate between groups on these datasets leading to systematic grouping errors. This failure mode of RF is exacerbated on Tetrominoes leading to very poor ARI scores (in Table 1).
>
> In contrast, we see that our model is capable of disambiguating the separate entities in groups of two or three tetris blocks with same color and texture (row 2, 3 in Figure 7) or two hearts (row 2 in Figure 4 and row 1 in Figure 8) or three squares (row ) or two red balls (row 2 in Figure 9) or two blue cubes (row 3 in Figure 9).
>
> end of part 1/3

---

> ### Author Response · Authors · 2024-08-06
> **Rebuttal Response -- (part 2/3)**
>
> ``` It is understandable that the test loss is lower for the model without bottlenecks since it has a greater number of parameters. ```
>
> Actually, the reviewer's claim is incorrect. The confusion is natural and our fault because ordinarily when discussing 'bottlenecks', spatial resolution and number of parameters are confounded. However, because we were concerned about this, our experiments actually ensure that both ‘SynCx’ and ‘SynCx w/o bottleneck’ have the same number of parameters. Instead we only vary the spatial resolution of feature maps. So the bottleneck is in spatial resolution but not in the number of parameters. We must be more explicit on this point in our paper; thanks to the reviewer for highlighting the natural confusion. We will adjust the corresponding text to communicate this better.
>
> ``` The importance of bottlenecks, in my opinion, originates due to the effective receptive field sizes in these layers and has nothing to do with complex-valued units or synchrony per se. It is unclear what this analysis adds to the manuscript. ```
>
> Bottlenecks ( using convolution with stride > 1) in representational capacity are crucial for a fully convolutional autoencoder because they force it to learn a compressed representation of the input in the latent layers. To test the reviewer’s hypothesis about effective receptive field size being the mechanism that drives phase synchronization, we ran another control experiment where we increased the receptive field size by increasing the kernel size of the convolution filters in our model while using a stride of 1 (no bottlenecks).
>
> | Kernel size | MSE                | ARI          |
> |-------------|--------------------|--------------|
> | k=3         | 6.29e-5 +-9.00e-5  | 0.10 +- 0.06 |
> | k=4         | 1.27e-3 +- 6.12e-5 | 0.67 +- 0.04 |
> | k=5         | 2.97e-6 +- 1.04e-6 | 0.52 +- 0.03 |
> | k=6         | 4.73e-2 +- 2.21e-6 | 0.00 +- 0.00 |
>
> The Table above shows results for our model which uses increasingly larger kernel sizes (3, 4, 5 and 6) and stride of 1 (no bottleneck). We can see that even when the kernel sizes are increased without having any bottlenecks it does not lead to increasingly better phase synchronization towards objects. The presence of representational bottlenecks (smaller spatial resolution of feature maps) using convolutions with stride > 1 leads to much better phase synchronization behavior in our models compared to the results shown above.
>
> ``` From the results presented, it seems like there is diminishing returns after just 2 iterations? If that's the case, then wouldn't this approach be very comparable to feedforward methods? Moreover, can the authors train SynCx on a large number of timesteps (say 10) and observe the dynamics of convergence in the phases during test time? Claims about "iterative constraint propagation" needs further backing. Perhaps these datasets are too easy? ```
>
> Yes, we agree with the reviewer's hypothesis. As supporting evidence, we find that as the visual complexity of the data set increases, the number of iterations to converge to the best decomposition increases (e.g., 3 iterations for Tetrominoes and dSprites, but 4 for CLEVR). We were also surprised that convergence would be so quick and indeed, any recurrent constraint propagation algorithm can be implemented as a feedforward (unrolled) model. We describe the process as constraint propagation to provide an intuition about the nature of the computation. And we agree with the reviewer that as we move to even more complex data sets, the number of iterations will increase further. Lastly, please note that there is still a way to go for these models to achieve perfect grouping scores ARI: 1.0 on these datasets especially on CLEVR so we believe the set of datasets picked are an appropriately difficult benchmark suite to test these synchrony-based models.
>
> ``` I found it hard to grasp the take away of this section. The Von-Mises distribution (with mean 0, and concentration 1) biases most of the units in the initialization to have similar phases. The random initialization performed pretty poorly. What does this suggest though computationally? ```
>
> This suggests that it is important to tune the variance of the distribution used to sample the “noisy” phase map. The Von-Mises distribution that we use is simply the circular analogue of the normal distribution which is a reasonable choice. This section provides practical recommendations for initialization schemes for the initial “noisy” phase map. Since the adoption and training of complex-valued autoencoders is still largely underexplored (non-standard) in the deep learning research community compared to the default deep learning models that use real-valued weights/activations. We believe it’s important to detail the key engineering choices to facilitate future users to train such autoencoders well on new benchmarks.
>
> end of part 2/3

---

> ### Author Response · Authors · 2024-08-06
> **Rebuttal Response -- (part 3/3)**
>
> ``` Figure 2 is slightly misleading in its depiction. The magnitude of the output \mu_z^1 is shown to be the same as \mu_x. This derails the reader from understanding the flow of information across timesteps. Notationally, denoting time as a superscript and outputs as z’s is confusing from a reader’s perspective. A clearer version could be something along the lines of \hat{x}(t) = \mu_{\hat{x}}(t) \odot \exp{i \phi_{\hat{x}} (t)} ```
>
> We thank the reviewer for the suggestions for improving Figure 2. The model diagram (and the corresponding input/output images) are meant to aid reviewers' understanding of the model and are therefore an approximation of the process. We will replace the output magnitude image with a reconstructed image instead. We thank the reviewers’ suggestion for a revised notation for the complex input $x(t)$. Please note, the magnitude component is independent of the iteration and is always clamped to the input image ($\mu_x$). Accordingly, we can rewrite the expression as $ x(n) = \mu_x * e^{i \phi_{\hat{x}(n)}}$. Lastly, we apologize for the typo in line 159: clamped to the image ($x'$) -> clamped to the image ($\mu_x$).
>
> Minor (typos):
> We thank the reviewer for these corrections and will revise them.
>
> end of part 3/3

---

> > ### Comment · Reviewer_Ta5A · 2024-08-12
> > **rebuttal feedback**
> >
> > Dear authors,
> >
> > Thanks for your detailed rebuttal.
> >
> > 1. It is commendable that the authors openly discuss the shortcomings of their method. Adding a discussion to address points such as the loss of phase information at the outputs (for 1-D phase reps), effect of time unrolling as a function of dataset complexity, etc. will be very useful. Particularly on the issue of iterations to converge: it's interesting that if the datasets are challenging enough, then why doesn't the model utilize recurrence more?
> >
> > 1. I wish to clarify the point about receptive field sizes. In traditional convolutional models, the effective receptive field sizes increase because of "pooling" operations. Here, the authors leverage strides instead (but the overall idea is the same). Increasing the kernel sizes will only increase the effective receptive field sizes by a few pixels and not, for example, by a factor of two. So, while the results of the new experiment the authors present in the rebuttal is good to know, my original point still stands I believe.
> >
> > 1. Thanks for clarifying the points about initialization, bottlenecks, and iterations. My genuine suggestion is that these sections do not contribute to the manuscript in a meaningful way. The introduction of the paper was geared toward perceptual grouping and gestalt psychology -- and I believe there are a lot of scientific insights to unpack here. The sections as they are now are more about implementational tricks and suggestions. I would urge the authors to use this space wisely.
> >
> > I am updating my score to 5. I would really love for this paper to be in a better shape for me to recommend stronger acceptance. Good luck to the authors!

---

> > > ### Author Response · Authors · 2024-08-13
> > > **rebuttal discussion**
> > >
> > > Thank you for the feedback and encouragement. We will comment on the loss of phase information in outputs. Regarding the increased size of convolutional kernels by a factor due to pooling or striding, in the experiments shown above, with k=6 shows a 2x increase in receptive field size. We experimented with even larger receptive fields (k=7, 8 and 9 -> 3x increase in size) for convolutions using stride=1 but they show no signs of phase synchronization (ARI score of 0.0). But we understand the point you make as well. On the usage of space in the paper across sections, we’ve tried our best to maintain a balance between elucidating computational principles and engineering methodologies necessary to realize phase synchrony in complex-valued networks under practical settings. We will continue to improve our presentation form.

---

### Author Response · Authors · 2024-08-06
**General Response to all Reviewers**

We have provided individual responses to each reviewer's questions/comments. We have not made any changes to the PDF of the paper to ensure the ease of referring to the line numbers, Figure/Table numbers are as in the original version. We’ve provided additional results (Tables) as part of the individual rebuttal responses.

Consolidated list of notable changes in the updated manuscript based on reviewer feedback.

- Commenting on the use of latent v/s output phases for grouping and limitations behind this choice.
- Additional control experiment that shows increased receptive field size of convolutions does not lead to increased phase synchronization towards objects.
- Clarify that bottlenecks in SynCx, we refer to bottlenecks in feature map resolution and not model parameters. Both SynCx and SynCx w/o bottlenecks (in Table 2) have equal number of model parameters.
- Changes in Figure 2 (model diagram) to edit output magnitude maps shown to not be exactly identical to the input image.
- Additional experiment showing how well the grouping performance of SynCx ‘extrapolates’ when using greater iterations at test time compared to training.
- Addition of object discovery results from SlotAttention baseline to Table 1 and commenting on the performance gap between SoTA slot-based and synchrony-based models.
- Highlight the parameter and training efficiency gains of our model against baselines. Addition of Tables comparing the model capacity and training times of our model against baselines which shows that our model is between 6-23x more parameter efficient than all the baselines and that our model also takes less time to train compared to baselines.
- Additional result in Table 4 showing the case for a constant zero phase map initialization as done in CAE.
- Correct citation for modReLU activation.

We thank all reviewers for their constructive feedback that helped improve our work. We’re happy to continue engaging with all reviewers to resolve any remaining concerns.

---

### Author Rebuttal · Authors · 2024-08-06

Please find attached a PDF containing samples of the UMAP v/s t-SNE comparison for dimensionality reduction in our phase map visualization process.

---

### Decision · Program_Chairs · 2024-09-25

**Decision:**

Accept (poster)

**Comment:**

This paper proposes a novel approach for unsupervised object discovery using a recurrent model with complex-valued weights that is validated against several standard benchmarks. The use of recurrence and complex-valued operations makes this paper potentially interesting to neuroscientists and cognitive scientists who seek to understand how temporal dynamics of neural populations might lead to synchrony-based operations that support cognition. Eventually, insights about binding-by-synchrony could also support the development of a new class of deep neural networks for complex sensory and cognitive tasks.

These factors led to three of the reviewers giving this paper above-threshold scores. However, there was also sentiment that the contributions of this paper were (1) not significant, and (2) not clearly translatable to neuroscience, cognitive science, or AI. In terms of the contributions of the paper, one of the reviewers noted that the effect of recurrence in this model is not clear. Indeed, the authors wrote in their introduction that the recurrence of the model is a key advancement beyond prior works, but the impact of recurrent steps of processing is minimal (performance caps out after 2 or 3 iterations). Moreover, the impact of complex valued units should in principle be more interpretable representations by multiplexing different computations in the phase vs. magnitudes of units. However, the authors note that directly interpreting phase information can be challenging at the output of the model. I will also note that the fact that the authors did not provide source code with their submission makes it more challenging to directly evaluate claims of interpretability or the effect of recurrence.

Overall, this is a paper with interesting ideas and potential impact by advancing the state-of-the-art on unsupervised object discovery, albeit on toy datasets. There is however a very limited impact on the cognitive sciences or AI more broadly, so I recommend the authors dial back that rhetoric. This is a true borderline paper that the reviewers were split on and could not resolve their issues despite a vigorous discussion period, but given two of the reviewers saw high potential impact, I lean towards accepting this.